



# A multi-physics Eulerian framework for long-term contrail evolution

Amin Jafarimoghaddam[1] and Manuel Soler[1]

[1]Department of Aerospace Engineering, Universidad Carlos III de Madrid, Avenida de la Universidad 30, Leganés, 28911 Madrid, Spain

**Correspondence:** Amin Jafarimoghaddam (ajafarim@pa.uc3m.es)

**Abstract.** Condensation trails (contrails) are increasingly recognized as a major contributor to aviation-induced atmospheric warming, rivaling the impact of carbon dioxide. Mitigating their climate effects requires accurate and computationally efficient models to inform avoidance strategies. Contrails evolve through distinct stages, from formation and rapid growth to dissipation or transition into cirrus clouds, where the latter phase critically determines their radiative forcing. This long-term evolution is primarily driven by advection-diffusion processes coupled with ice-particle growth dynamics. We propose a new multi-physics Eulerian framework for long-term contrail simulations, integrating underexplored or previously neglected factors, including spatiotemporal wind variability; nonlinear diffusion coefficients accounting for potential diffusion-blocking mechanisms; a novel multiphase theoretical model for the bulk settling velocity of ice particles; and ice-crystal habit dynamics. The Eulerian model is solved using a recently proposed discretization approach to enhance both accuracy and computational efficiency. Additionally, the Eulerian model introduces several theoretical, adjustable parameters that can be calibrated using ground-truth data to optimize the built-in nonlinear advection–diffusion equations (ADEs). We further demonstrate that the governing nonlinear ADEs admit dimensional separability under suitable assumptions, making the multi-physics Eulerian model particularly promising for large-scale simulations of contrail plumes and, ultimately, their associated radiative forcing.

## 1 Introduction

Contrails, visible ice-crystal clouds generated by aircraft engine exhaust, are now suspected to be one of the leading anthropogenic contributors to radiative forcing, with a warming effect at least comparable to that of $CO_2$ emissions, albeit with a high level of uncertainty (Engberg et al., 2025). After rapid nucleation process and initial growth phase, contrails may persist for hours, undergoing complex interactions between ambient winds, turbulent mixing, and microphysical growth processes before dissipating or transitioning into cirrus clouds. Capturing these evolution stages is essential for accurate estimation of contrail-induced climate impacts and for the development of mitigation strategies. Modeling approaches typically fall into the following categories: (i) high-resolution simulations of individual contrails; (ii) Lagrangian plume models that follow contrail segments with prescribed microphysics; and (iii) parameterized schemes in global models (Paoli and Shariff, 2016).



Although contrail dynamics have been the subject of extensive research, no existing numerical framework has yet demonstrated sufficient accuracy to reproduce contrail evolution over climatological timescales, nor to rigorously quantify the associated radiative forcing (Akhtar Martinez and Jarrett, 2024). Most existing models lack experimental validation and often rely on oversimplified macro- and micro-scale physics. Combined with limitations in the computational framework, this has led to inconsistent results across current contrail models (Akhtar Martínez et al.,

30 2025).

In this research, we present a unified Eulerian framework for contrail evolution that rigorously couples macro- and microphysical processes within a single computational domain. Macro-scale dynamics are described by moment equations derived from the Population Balance Equation (PBE) (Ramkrishna, 2000), which include a nonlinear diffusion term representing the (possible) diffusion-blocking mechanisms. Microphysics are represented by Eulerian

field equations obtained by translating the Lagrangian growth kinetics of individual ice crystals into spatially and temporally resolved moment fields. These microphysical fields are further coupled to a habit-dynamics field equation that solves shape-evolution, enabling a continuous representation of ice-crystal geometry throughout the contrail life cycle. Within this framework, we distinguish the particle-scale terminal velocity (function of crystal mass and projected area-known as Stokes formula) from the ensemble-scale bulk settling velocity. By performing an analysis

of the multi-phase flow equations under high turbulent mixing, we derive a first-order Burgers'-type equation that accounts for the collective bulk settling velocity.

In the following, we present a review of existing contrail models and simulations, emphasizing their key characteristics such as simulation scale, treatment of macro- and microphysical processes, model complexity, and computational framework. We also highlight the significance of the new elements introduced in this study, especially, ice crystal

habit dynamics and bulk settling velocity, integrated within a consistent Eulerian computational approach.

The Contrail Cirrus Prediction model (CoCiP) (Schumann, 2012) simulates contrail segments along aircraft flight paths within a Lagrangian Gaussian plume framework. Its wake vortex sub-model initializes the plume in the diffusion regime, assuming Gaussian spatial distributions of water vapor and ice water content. The resulting elliptical plume evolves under wind shear and diffusion, while its centroid descends due to ice crystal sedimentation. Ice microphysics

are represented by two evolving parameters: the total number of ice crystals (assumed monodisperse) and the total ice mass, which is constrained by the Schmidt–Appleman criterion (Schumann et al., 1995; Gierens et al., 2003; Seinfeld and Pandis, 2016) to capture growth and sublimation.

Although CoCiP is well suited for large-scale simulations and long-term contrail evolution, it omits several important processes. These include consistent coupling of advection and diffusion in the Lagrangian framework,

realistic treatment of bulk settling velocity beyond individual Stokes terminal velocities, explicit representation of particle size polydispersity (or grid-based sensitivity analysis of monodisperse assumption), and the evolution of ice crystal habits. Its treatment of short-term contrail development prior to the diffusion regime (Paoli and Shariff, 2016) is also limited, lacking detailed chemistry, aerosol interactions beyond ice mass, and ice nucleation dynamics. Similar



limitations apply to the Ames Contrail Simulation Model (ACSM) (Li et al., 2023), which is based on nearly the
same theoretical framework as CoCiP.

Aircraft Plume Chemistry, Emissions, and Microphysics Model (APCEMM) (Fritz et al., 2020) is a Lagrangian
plume-scaling model that explicitly simulates the two-dimensional cross-sectional evolution of aircraft exhaust
plumes. It applies the Schmidt–Appleman criterion to determine contrail formation and subsequently models the
evolution of vapor and particles through detailed microphysics and chemistry. APCEMM utilizes binned (sectional)
aerosol and ice microphysics alongside two-dimensional advection and diffusion processes to capture the contrail
cross-section dynamics. The model also incorporates a unified chemical mechanism encompassing both gas-phase and
heterogeneous reactions, accounting for exhaust species such as $NO_x$ and soot, and their influence on plume behavior.
Contrail ice naturally forms in the simulation once water vapor supersaturates on soot particles; thereafter, the ice is
transported and dispersed by ambient wind shear and gravitational settling. Compared to CoCiP, APCEMM is more
computationally intensive, and in practice, it typically predicts longer contrail lifetimes and stronger radiative forcing
than CoCiP under similar conditions (Xu et al., 2023; Xu, 2024). While APCEMM advances contrail representation,
it still shares some of CoCiP's limitations in the long-term diffusion regime.

In high-fidelity contrail models, both RANS and LES are used to simulate the jet and near-wake stages of single
contrails. For example, Unterstrasser et al. have conducted key LES studies on the transition from young contrails to
persistent contrail cirrus, including parametric analyses of contrail evolution (Unterstrasser and Gierens, 2010a, b)
and comparisons with natural cirrus, focusing on local-scale interactions (Unterstrasser et al., 2017a, b). Their work
links small-scale contrail microphysics to larger-scale cirrus dynamics, emphasizing factors that control contrail
lifetime and climate impact. Guignery et al. performed steady RANS simulations with a detailed microphysics module
on a 2D wing with engine injectors, simulating ice formation up to eight wing spans downstream (Guignery et al.,
2012). Likewise, Montreuil et al. ran 3D RANS on a realistic wing–body–engine configuration (the NASA CRM)
coupled with Eulerian soot–ice microphysics to capture contrail onset (Montreuil et al., 2018). Lewellen conducted
large-eddy simulations of the exhaust jet with bin-resolved microphysics to quantify ice nucleation and growth and
the resulting ice-number emission index (Lewellen, 2020). Bouhafid et al. simulated the jet regime with RANS and
then ran LES for the vortex/dissipation stages, using synthetic turbulence to transfer RANS-derived fluctuations into
the LES domain (Bouhafid et al., 2024).

In the following, we present concise observations on two microphysical simplifications common to contrail models.
**Habit development.** Persistent contrails and contrail cirrus are dominated by faceted, non-spherical ice crystals
that closely resemble those in natural cirrus clouds. Both in situ observations and remote sensing studies consistently
demonstrate that contrail ice particles deviate from spherical shapes. For instance, Yang et al. (Yang et al., 2010) report
that contrails and contrail-cirrus clouds are composed almost exclusively of nonspherical ice crystals. Polarimetric
lidar measurements further support this by revealing strong light depolarization signals indicative of non-spherical
particles. Similarly, CALIPSO satellite retrievals confirm that non-spherical ice dominates persistent contrails (IPCC,
1999). Iwabuchi et al. (Iwabuchi et al., 2012) find that depolarization ratios measured in contrails align well with





theoretical predictions for mixtures of randomly oriented, nonspherical ice habits. As contrails age and spread, larger
faceted crystals, including plates, columns, and rosettes, become increasingly prevalent. Complex crystal shapes such
as irregular bullet rosettes have also been observed.

    In summary, long-term contrails rapidly evolve into contrail cirrus composed of faceted, nonspherical ice crystals,
similar to natural upper-tropospheric cirrus. Concurrently, the effective radii of ice crystals increase with contrail
age. Recent reviews indicate that, in older contrails (age $> 120$ s), most ice crystals have effective radii ranging
from 10 to 150 $\mu$m (Krämer et al., 2020; Wolf et al., 2023). Field campaigns have also documented crystals reaching
several hundred microns in size (Iwabuchi et al., 2012). Specifically, once contrail ice crystals exceed a critical size
(typically around $5 - 10\mu$m radius), they experience microphysical behaviors analogous to those in natural cirrus
clouds. Empirical evidence from in situ sampling at approximately $-61°$C shows that contrail ice crystals with
effective radii larger than about 10 $\mu$m predominantly exhibit habits characteristic of natural cirrus: roughly 75%
hexagonal plates, 20% columns, and a smaller fraction of triangular plates. These habits emerge irrespective of the
initial nucleation mechanism. Therefore, mature contrail cirrus transition into the same microphysical regime as
natural cirrus, where habit classification based on temperature and supersaturation is appropriate (Yang et al.,
2010). However, the dominant ice crystal habits and their variation with contrail age and crystal size remain poorly
understood (Kärcher, 2018).

**Bulk settling velocity of ice particles.** The terminal velocity of an isolated, spherical particle (e.g. as estimated
from Stokes' law) is often used in long-term contrail models, but it does not capture collective, multiphase effects
that determine the effective downward speed of an ensemble of crystals, particularly for contrail models that neglect
explicit fluid–solid coupling. The *bulk* settling velocity denotes the ensemble-averaged settling behaviour of particles
embedded in a carrier fluid and thus inherently reflects particle–fluid and particle–particle interactions as well as
turbulence modulation. Multiphase simulations and laboratory experiments routinely document substantial departures
of ensemble settling from single-particle terminal velocities (Stout et al., 1995; Chen et al., 2020; Akutina et al., 2020;
Fornari et al., 2016), driven by processes such as *loitering*, and *preferential sweeping*.

    In practical contrail modeling it is common to retard vertical loss by imposing small ambient updrafts or by
accounting for the plume's initial buoyancy (stemming from the exhaust–ambient temperature contrast). These
expedients compensate, in part, for relatively large Stokes-based settling rates, especially for larger crystals. Moreover,
buoyancy is inherently short-lived (typically, minutes to $\mathcal{O}(1)$ h) and ambient updrafts are highly variable in space
and time, so these compensations are not universally justified.

    Accordingly, we argue that an approach which replaces isolated-particle Stokes formulas by an ensemble (bulk)
settling description coupled with crystal habit dynamics (shape-dependent drag and growth) captures physically
relevant processes missing from current long-term contrail models. Our numerical experiments indicate that including
ensemble settling effects together with habit dynamics reproduces plume retention characteristics similar to those
obtained by *ad hoc* updraft or buoyancy corrections, without explicitly imposing such compensating flows. Therefore,





although buoyancy and ambient updrafts can play a role, their typical magnitudes and time scales in the highly
variable UTLS should (perhaps) be re-evaluated before being prescribed in contrail parameterizations.

In this work, we develop a multi-physics Eulerian framework for long-term contrail evolution that retains two
moments of the PBE, ensuring rigorous conservation of particle number and mass. The model incorporates: (i)
spatiotemporally variable, nonlinear diffusion coefficients to capture the possible diffusion-blocking effects; (ii) a
new theoretical settling-velocity formulation that accounts for bulk settling velocity in turbulent flows; (iii) a
tracking field equation for ice particle habit dynamics along with their growth and settling mechanisms; and (iv)

an advanced discretization approach (Jafarimoghaddam et al., 2025) that enhances computational efficiency and
accuracy over standard solvers. In addition, we demonstrate that, under mild assumptions, the governing equations
exhibit separability, making the model particularly well-suited for large-scale simulations with a favorable balance
between accuracy and computational cost. By coupling detailed microphysical growth dynamics with a robust
moment-based advection-diffusion equation system in a consistent Eulerian setting, the proposed framework offers a

scalable and physically grounded tool for predicting long-term contrail evolution and their radiative impacts under
realistic atmospheric conditions.

## 2   Contrail Evolutionary Stages and and Assumptions in Long-Term Simulations

Modeling the advection–diffusion dynamics that govern the growth and transport of ice particles is fundamental
to understanding a wide array of atmospheric processes, most notably the formation and persistence of aircraft

contrails. Two primary paradigms have been employed. In the Lagrangian framework, one follows individual particle
trajectories through the flow field (e.g. (Sundqvist, 2003)). By contrast, the Eulerian approach describes the evolution
of particle-size (or mass) distributions via the Population Balance Equation (PBE) (Breit et al., 2025), often cast as
a coupled system of advection–diffusion equations (ADE) with suitable closure assumptions (e.g. (Pruppacher and
Klett, 1997)). In this study, we focus on the **long-term evolution** regime (after the jet-induced wake has decayed)

and introduce a novel Eulerian formulation that conserves both particle number and mass by retaining the zeroth
and first moments of the PBE.

**Stages of Contrail Evolution.** As illustrated in Fig. 1, contrail development progresses through three to four
overlapping regimes (Paoli and Shariff, 2016).

1. **Jet/vortex-regime (vortex roll-up) and nucleation (seconds):** Immediately downstream of the engine

exit, extremely high supersaturations drive rapid homogeneous or heterogeneous nucleation of very small ice crystals.
Hydrodynamic shear and coherent vortex structures dominate their dispersion and early collision/coalescence behavior.

2. **Intermediate vortex wake descent/break up (minutes):** As the strong jet vortices break down, crystals
remain small and are mixed by decaying turbulent eddies. Temperature and humidity start to relax toward ambient
values, but settling effects are still minor compared to turbulent mixing.





3. **Long-term diffusion and habit development (tens of minutes to hours):** Once the jet-induced turbulence has dissipated, only the ambient wind and residual turbulence remain. In this regime, two competing inertial-turbulence mechanisms, loitering (enhanced drag from small-eddy sampling slows settling) and preferential sweeping (partial decoupling drives larger crystals into downward-moving regions), govern the net particle descent. Our Eulerian model applies precisely in this stage, coupling the ice-particle velocity (ambient wind plus turbulence-modified settling) with microphysical source terms and diffusion closures. Notably, the net radiative forcing attributable to contrails is overwhelmingly determined during this long-term, contrail-cirrus stage.

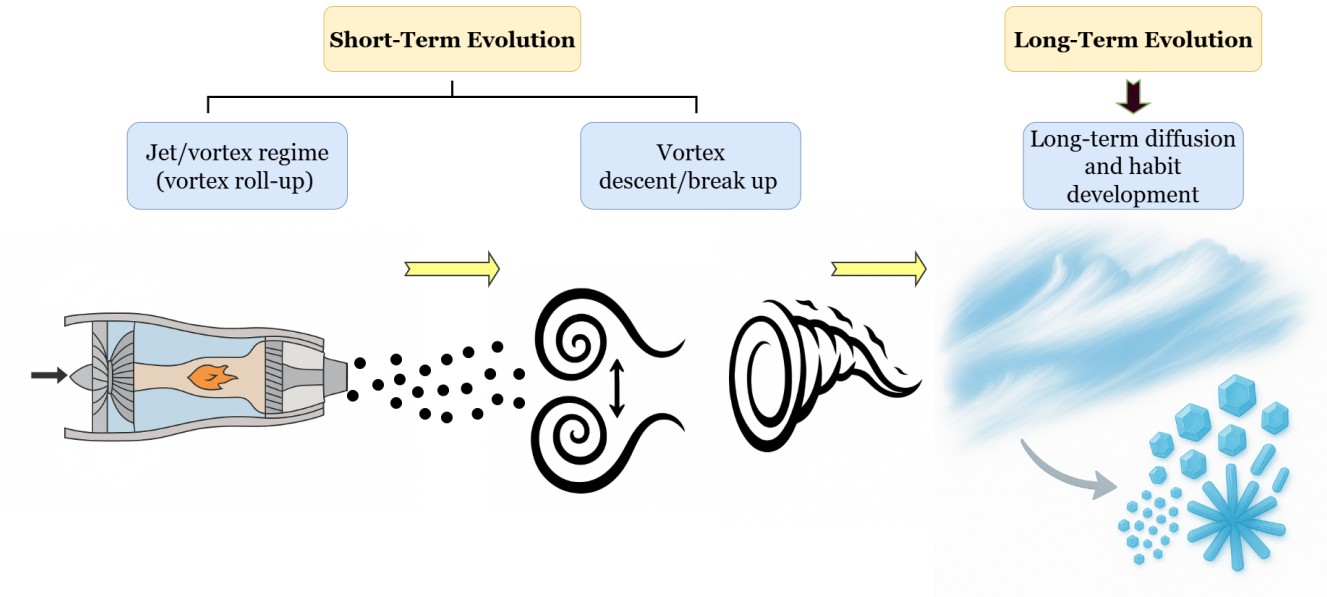

**Figure 1.** Schematic diagram of the short-term/long-term stages of contrail evolution

## 2.1    Modeling Assumptions

The development of the present model relies on the following assumptions.

First, post-formation nucleation is considered negligible. Following the initial rapid nucleation phase during contrail formation, no significant subsequent nucleation events are assumed to occur. As a result, the ice-particle number concentration remains conserved in the absence of external source/sink terms such as aggregation process (Pruppacher and Klett, 1997; Sundqvist, 2003).

Second, we assume the dominance of ambient wind-driven advection. Transient aerodynamic effects, such as wake vortices and aircraft-induced jets, are regarded as short-term phenomena. Consequently, the long-term advection of contrail particles is driven primarily by the ambient wind field and the gravitational settling velocity of the particles.

Third, we adopt a spheroidal approximation for ice-particle geometry. This approach, consistent with the modeling framework of Chen and Lamb (Chen and Lamb, 1994), and Nelson and Baker (Nelson and Baker, 1996) allows the





treatment of complex hexagonal habits as spheroids, thereby generalizing the mathematical representation of growth and settling dynamics.

Finally, each computational cell is characterized by an average ice-particle mass, number concentration, mass concentration, and shape index, thereby assuming a locally monodisperse particle distribution through the use of the Kronecker delta function in the population balance equation. Nevertheless, the grid dependency of the plume in the sensitive vertical dimension was assessed, and a fine mesh resolution was selected at which the field equations were convergent, thus supporting the applicability of the monodispersity assumption.

## 3    Multi-Physics Equations for Long-Term Contrail Evolution

### 3.1    Governing Equations for Bulk Motion

Based on the above assumptions, we begin modeling the evolution of persistent contrails by the following system of coupled advection–diffusion equations (see Appendix B):

$$
\frac{\partial c_N}{\partial t} + \nabla \cdot \left( \mathbf{v}_p \, c_N \right) = \nabla \cdot \left( \tilde{\mathcal{D}} \, \nabla c_N \right) + S_{c_N},
$$
$$
\frac{\partial c_M}{\partial t} + \nabla \cdot \left( \mathbf{v}_p \, c_M \right) = \nabla \cdot \left( \tilde{\mathcal{D}} \, \nabla c_M \right) + c_N \, \rho_{\mathrm{dep}} \, f_{\mathcal{V}} + S_{c_M}. \tag{1}
$$

In the equations above, $t$ is time, $\mathbf{x} := (x, y, z)$ is the system coordinate in a Cartesian framework, $c_N := c_N(\mathbf{x}, t)$ and $c_M := c_M(\mathbf{x}, t)$ represent the ice-particle number and mass concentrations, respectively. The bulk velocity of the ice particles is given by $\mathbf{v}_p := \mathbf{v}_p(\mathbf{x}, t)$, and the nonlinear stochastic isotropic diffusion coefficient matrix is denoted by $\tilde{\mathcal{D}} := \tilde{\mathcal{D}}(\mathbf{x}, t, c_N)$ (see Appendix G). Moreover, $S_{c_N} := S_{c_N}(\mathbf{x}, t)$ and $S_{c_M} := S_{c_M}(\mathbf{x}, t)$ represent the source terms for the ice-particle number and mass concentrations, respectively, due to additional release of ice particles. In the

present study the source/sink term is neglected (i.e., $S_c = 0$). However, for large-scale simulations, the term $S_c$ may be included in the governing transport equations to account for additional release of ice particles. Alternatively, for large-scale simulations, one may set $S_c = 0$ and adjust the initial conditions to represent a continuous release of ice particles. Finally, $f_{\mathcal{V}} := f_{\mathcal{V}}(a, \phi, \mathbf{x}, t)$ represents the volumetric growth rate of individual particles, $\rho_{\mathrm{dep}}$ denotes the effective deposition density, where $a$ is the ice particle's equatorial radius and $\phi$ is the ice-particle shape index, which

will be elaborated in the next section.

### 3.2    Governing Equations for Individual Ice Particles' Dynamics

Modeling persistent contrails also requires microphysical equations that describe the evolution of individual ice particles' mass and shape. Accurate parameterization of contrail single-scattering properties, and thus reliable simulation of contrail and contrail-cirrus radiative forcing, requires explicit representation of the crystal nonsphericity

(Yang et al., 2010). Employing the habit dynamic framework by Chen and Lamb (Chen and Lamb, 1994), and Nelson and Baker (Nelson and Baker, 1996), the coupled ODEs describing the evolution of ice crystal mass and shape can





be written as:

$$\frac{dm}{dt} = \rho_{\text{dep}} \, f_{\mathcal{V}}(a, \phi, \mathbf{x}, t), \qquad \frac{d\phi}{d\mathcal{V}} = \frac{\Gamma^*(T, s_i) - 1}{\Gamma^*(T, s_i) + 2} \frac{\phi}{\mathcal{V}}, \qquad \mathcal{V} = \frac{4\pi}{3} \, a^3 \, \phi. \qquad (2)$$

In the above, $m$ is the individual ice particle mass, $\mathcal{V}$ corresponds to the volume of an individual ice particle, while $\phi$
represents the individual ice-particle shape index, defined as $\phi := \frac{c}{a}$, where $a$ is the equatorial radius and $c$ is the transverse radius; $\phi > 1$ indicates columnar (prolate) crystals and $\phi < 1$ denotes plate-like (oblate) crystals. $T := T(\mathbf{x}, t)$ is the spatiotemporal temperature field, and $s_i := s_i(\mathbf{x}, \mathbf{t})$ represents the spatiotemporal background/ambient ice supersaturation field. In addition, $\Gamma^*$ is known as Inherent Growth Factor (IGF), and $(f_{\mathcal{V}} := f_{\mathcal{V}}(a, \phi, \mathbf{x}, t))$ denotes the volumetric growth rate of individual ice particles (see Appendix E, and F)[1].

For completeness, we note that laboratory evidence suggests the aspect ratio of ice crystals remains constant during sublimation. This is attributed to the uniformity of vapor density along the crystal surface, which results in shape preservation throughout the sublimation process (Nelson and Knight, 1998; Magee et al., 2014; Ham, 1959).

Notably, since our objective is to develop an entirely Eulerian framework for modeling persistent contrails, we also translate the individual ice-particle dynamics into an appropriate Eulerian representation. In other words, the
Eulerian framework requires that the mass and shape dynamics, $m(t)$ and $\phi(t)$, be reformulated as field equations $m(\mathbf{x}, t)$ and $\phi(\mathbf{x}, t)$ through additional ADEs. However, the field quantity $m(\mathbf{x}, t)$ can be obtained from the identity $c_M := m c_N$, implying that solving for $c_M$ also yields $m(\mathbf{x}, t)$. Nevertheless, we provide an explicit ADE for $m(\mathbf{x}, t)$ both for completeness of the Eulerian framework and as an alternative to the ADE for $c_M$.

We also highlight that the spatial dependence of $m$ and $\phi$ is significant since temperature and ice supersaturation
are spatiotemporal fields, i.e., $T(\mathbf{x}, t)$, and $s_i := s_i(\mathbf{x}, \mathbf{t})$. More specifically, if one were to assume $T = T(t)$, and $s_i = s_i(t)$, then one would have $m = m(t)$ and $\phi = \phi(t)$, implying that the Eulerian framework would collapse to a Lagrangian one.

### 3.3 The Proposed Eulerian Framework for Persistent Contrails

As discussed, we can obtain an ADE accounting for the evolution the mass field $m(\mathbf{x}, t)$ directly (see the derivation
details in Appendix B)[2]. However, it is not possible to directly formulate an ADE for particle shape evolution in the same sense as for mass concentration. Although a particle entering a control volume transports both its mass and specific shape, only mass is a conserved scalar obeying continuity laws. In contrast, particle shape evolves through

---

[1]Notably, parameterizations of ice-vapor growth exert a first-order control on cold-cloud behavior (Gierens et al., 2003; Avramov and Harrington, 2010). While laboratory and theoretical advances have yielded useful, approximate growth models, the detailed physical processes governing ice-crystal development remain incompletely characterized. In addition, the ice crystal quantities measured at different temperatures cannot be directly incorporated into the capacitance model, a formulation that nonetheless underlies many atmospheric ice-growth simulations (see e.g., (Libbrecht, 2003; Nelson and Knight, 1998)). This inconsistency highlights a persistent gap between process-oriented laboratory observations and the bulk parameterization employed in large-scale atmospheric models.

[2]It is noteworthy that the equation governing $m(\mathbf{x}, t)$ can also be derived directly from the quotient rule $c_M := m c_N$ by defining the net inlet flux into the control volume as $\mathbf{J}_{in} := m(\mathbf{x}, t) c_N(\mathbf{x}, t)$, and the net outlet flux expressed as $\mathbf{J}_{out} = \mathbf{J}_{in} + \frac{\partial}{\partial \mathbf{x}_i} \mathbf{J}_{in}$. The derivation is accomplished on using the conservation of the number concentration as the closure equation.





micro-physical processes that fundamentally differ from diffusion. Therefore, in the Eulerian framework for particle shape evolution, only an advection term is incorporated to ensure the presence of the shape index $\phi(\mathbf{x}, t)$ in all regions where particles are present, followed by a term describing its micro-physical evolution that determines the rate at which particle shapes change. Therefore, the Eulerian equations governing the long-term evolution of contrails are presented as:

$$\frac{\partial c_N}{\partial t} + \nabla \cdot (\mathbf{v}_p c_N) = \nabla \cdot (\tilde{\mathcal{D}} \nabla c_N),$$

$$\frac{\partial m}{\partial t} + \mathbf{v}_p \cdot \nabla m = \nabla \cdot (\tilde{\mathcal{D}} \nabla m) + \frac{2\tilde{\mathcal{D}}}{c} \nabla m \cdot \nabla c_N + \rho_{\text{dep}} f_{\mathcal{V}},$$

$$\frac{\partial \phi}{\partial t} + \mathbf{v}_p \cdot \nabla \phi = \frac{\Gamma^*(T, s_i) - 1}{\Gamma^*(T, s_i) + 2} \frac{\phi}{\mathcal{V}} f_{\mathcal{V}},$$

$$\mathcal{V} = \frac{4\pi}{3} a^3 \phi. \tag{3}$$

In the following section, we present the theory and formulas for the remaining building block of the Eulerian model, which includes the ice particle velocity $\mathbf{v}_p$.

## 4 Ice Particle Velocity $\mathbf{v}_p$

The total ice-particle bulk velocity field $\mathbf{v}_p(\mathbf{x}, t)$ is defined as:

$$\mathbf{v}_p = \mathbf{v}_{slp} + \underbrace{(w_x, w_y, w_z)^\top}_{\substack{\text{background velocity} \\ \text{(wind components)}}} \approx (0, 0, v_s)^\top + \underbrace{(w_x, w_y, w_z)^\top}_{\text{background velocity}} = (w_x, w_y, w_z + v_s)^\top. \tag{4}$$

In Eq. (B2), $\mathbf{v}_{slp}$ represents the bulk slip velocity of the particle phase within a turbulently mixing fluid, while $v_s$ denotes the bulk settling velocity. In the subsequent section, we develop a new model for $v_s$ derived from a rigorous analysis and reduction of the multiphase flow equations, incorporating an Euler–Euler framework for interphase momentum exchange and particle transport. Given that the mean vertical wind component $w_z$ is negligible, subgrid-scale vertical fluctuations can be embedded into the definition of $v_s$.

### 4.1 Wind-Field Model

Exact Navier–Stokes solutions at contrail scales (for resolving wind) are infeasible; accordingly we superpose synthetic turbulence onto ERA5-derived mean winds, while modeling the mean field with a composite inviscid representation. In principle the atmosphere is a turbulent, incompressible fluid governed by the Navier–Stokes equations, and the resulting eddy field spans scales from synoptic down to the inertial subrange, making a full-scale direct solution computationally prohibitive; nevertheless the incompressibility constraint $\nabla \cdot \mathbf{u} = 0$ holds at all scales. Because the velocity components are coupled through the momentum equations, a physically consistent, data-informed model should enforce incompressibility and, at large scales, approximate momentum balance in the inviscid limit. To this end





we represent the mean wind as a superposition of analytic, divergence-free primitives; uniform flow, regularized point vortices, potential dipoles, and regularized sources/sinks, each an exact solution of the incompressible Euler equations. The composite model therefore satisfies $\nabla\cdot\mathbf{u} = 0$ by construction and captures dominant large-scale patterns with a

small number of physically interpretable parameters.

Let $(x, y) \in \mathbb{R}^2$. We define the total wind-field as:

$$\mathbf{W}(x,y) = (w_x, w_y) = \underbrace{U_\infty\,\mathbf{e}_x + V_\infty\,\mathbf{e}_y}_{\text{uniform flow}} + \underbrace{\sum_{k=1}^{M_v} \mathbf{W}_k^{(v)}(x,y)}_{\text{vortex flow}} + \underbrace{\sum_{\ell=1}^{M_d} \mathbf{W}_\ell^{(d)}(x,y)}_{\text{dipole flow}} + \underbrace{\sum_{j=1}^{M_s} \mathbf{W}_j^{(s)}(x,y)}_{\text{source/sink flow}}. \tag{5}$$

where $M_v, M_d$, and $M_s$, are the number of introduced vortices, dipoles, and sources/sinks respectively (the primitives are specified in Appendix A). Specifically, for time-dependent reconstruction we fit the composite model (5) to each

instantaneous measurement. For snapshot $t_i$ the parameter vector $\mathbf{a}_i$ is obtained by:

$$\mathbf{a}_i = \arg\min_{\mathbf{a}} \mathcal{J}\big(\mathbf{W}_{\text{obs}}(\cdot, t_i), \mathbf{W}(\cdot; \mathbf{a})\big). \tag{6}$$

Each parameter component is then promoted to a continuous function of time and approximated by a polynomial, $a_j(t) \approx \sum_{k=0}^{n} c_{j,k}\, t^k, j = 1, \ldots, \dim(\mathbf{a})$, so that the time-dependent wind field reads $\mathbf{W}(x, y, t) = \mathbf{W}\big(x, y; \mathbf{a}(t)\big)$.

### 4.2   Bulk Settling Velocity $v_s$: A Low-Order Theoretical Model

Traditional approaches to particle settling rely on the terminal velocity derived for individual particles in unbounded domain using Stokes' law. However, in an Eulerian framework that describes the bulk concentration of particles (such as ice particles in contrails), the settling velocity must reflect the collective behavior of particles that are also subject to turbulent mixing[3]. In high-altitude regions, significant turbulent mixing implies that the ice particles initially 'hover' within the eddy-viscous layer while undergoing growth (provided that the growth conditions are met).

This motivates a model in which the effective settling velocity is determined not only by gravitational and drag forces but also by turbulent (or self-) diffusion. To differentiate notations, we use $v_{\mathbf{s}}$ for the particle effective/bulk settling velocity and $v_{\mathbf{ter}}$ for the terminal settling velocity, typically derived from Stokes' law (or alternative empirical formulas) for a single settling particle in an unbounded domain.

In turbulent flows, small particles can experience a phenomenon known as *loitering*. Due to their ability to be

readily carried by rapidly fluctuating eddies (sometimes referred to as the eddy-locking, analogous to behavior seen in nanofluids (Buongiorno, 2006; Jafarimoghaddam et al., 2021)), these particles sample a broad range of flow directions and, as a consequence, effectively experience enhanced nonlinear drag and may follow a longer trajectory to settle across a given vertical distance. This results in an average fall speed that is reduced relative to predictions based on quiescent conditions. A second mechanism, commonly referred to as *preferential sweeping*, occurs when particles

---

[3]The same comment applies to contrail models formulated within a Lagrangian framework without coupling to the surrounding fluid phase.





possess sufficient inertia to partially decouple from the turbulent eddies. They are centrifuged out of vortical cores and thus become concentrated in downward-moving regions of the flow, causing them to sample stronger downward velocity fluctuations and hence to acquire an enhanced mean settling velocity. The relative influence of these two mechanisms depends on the degree to which the particles are able to track turbulent fluctuations. Very small crystals tend to follow the turbulence closely, resulting in persistent agitation and reduced descent rates due to the *loitering*

effect. *Preferential sweeping* becomes significant when particles grow large enough, or turbulence weakens sufficiently, to allow them to slip out of the smallest vortices. In the specific case of contrail cirrus, where ice crystals remain very small under considerable atmospheric turbulence, the *loitering* mechanism is expected to dominate the early stages of evolution. Over time, as the particles grow by nearly an order of magnitude, they are expected to begin *sweeping* toward their terminal velocity (or even exceed it) typically when their Stokes number reaches $St = \mathcal{O}(1)$ or their

Galileo number reaches $Ga = \mathcal{O}(10^2\text{–}10^3)$, both of which are characteristic of larger particles (typically with radii greater than $50\text{ –}100\,\mu\text{m}$) (interested readers are referred to (Stout et al., 1995; Chen et al., 2020; Akutina et al., 2020; Fornari et al., 2016; Kaveh and Malcherek, 2024)).

In this section, we develop a low-order theoretical model that captures these effects; specifically, the initial *loitering* behavior decays asymptotically into *preferential sweeping*, thereby recovering the terminal velocity.

In turbulent flows, particularly under quasi-isotropic conditions, it is reasonable to assume that the turbulent diffusivity and the eddy viscosity are related by a turbulent Schmidt number, $Sc_t$, such that: $\tilde{\mathcal{D}} \approx \frac{\nu_t}{Sc_t}$. For many atmospheric and geophysical flows, the turbulent Schmidt number is of order unity. In our formulation, we adopt an effective turbulent viscosity $\nu_{\mathbf{t,ef}}$ that represents the 3D-averaged mixing. This is justified by the fact that the overall turbulent mixing experienced by the particles is inherently three-dimensional. Thus, we write: $\nu_{\mathbf{t,ef}} \approx \langle \tilde{\mathcal{D}}_{(x,y,z)} \rangle$,

accounting for the total mixing of the particle-laden flow. The directional diffusivities $\tilde{\mathcal{D}}_x, \tilde{\mathcal{D}}_y, \tilde{\mathcal{D}}_z$ vary with ambient turbulence and stratification; for far-wake, cruise-altitude contrails we adopt $\tilde{\mathcal{D}}_x = \tilde{\mathcal{D}}_y \in [10, 40]\ \text{m}^2\,\text{s}^{-1}$, and $\tilde{\mathcal{D}}_z \in [0.05, 0.50]\ \text{m}^2\,\text{s}^{-1}$ giving the arithmetic mean $\nu_{\mathbf{t,ef}} \in [6.68, 26.83]\ \text{m}^2\,\text{s}^{-1}$.

Beginning with the Euler-Euler framework for multiphase flows, where fluid and particulate phases are interpreted as continua, the momentum equation for the particle phase can be written as (see Appendix C):

$$\frac{\partial \mathbf{v}_{\text{slp}}}{\partial t} + (\mathbf{v}_{\text{slp}} \cdot \nabla)\mathbf{v}_{\text{slp}} = \nu_t \nabla^2 \mathbf{v}_{\text{slp}} + \boldsymbol{C}_f + \boldsymbol{C}_{c_M, c_N, f_{\mathcal{V}}}. \tag{7}$$

where $\boldsymbol{C}_f$ represents the coupling between $\mathbf{v}_{\text{slp}}$ and the fluid phase, and is expressed as $\boldsymbol{C}_f = -\frac{\nabla p}{\rho_p} + \mathbf{g} + \frac{f_d}{\rho_p}$; and $\boldsymbol{C}_{c_M, c_N, f_{\mathcal{V}}}$ captures the coupling of $\mathbf{v}_{\text{slp}}$ to the mass and number concentration fields, as well as to the growth term, and is given by $\boldsymbol{C}_{c_M, c_N, f_{\mathcal{V}}} = -\frac{\mathbf{v}_{\text{slp}}}{\epsilon_p} \left[ \nabla \cdot \left( \tilde{\mathcal{D}} \nabla \epsilon_p \right) + f_{\mathcal{V}} \left( \bar{m}(\mathbf{x},t), \mathbf{x}, t \right) c_N(\mathbf{x}, t) \right] + \nu_t \left[ \frac{1}{\epsilon_p} (\nabla \epsilon_p \cdot \nabla)\mathbf{v}_{\text{slp}} + \frac{1}{\epsilon_p} (\nabla \epsilon_p \cdot \nabla)^{\top} \mathbf{v}_{\text{slp}} \right]$.

Therefore, as detailed in (Appendix C), fully resolving the particle momentum equation requires additional closures

for the fluid phase, as well as for $f_d$, which represents the average drag force experienced by individual particles, distinct from the drag force on a single settling particle in an unbounded domain, and also depends critically on the specification of appropriate boundary and initial conditions.



The present model assumes that particles are taken to be initially at rest on a reference plane $z_{\mathbf{ref}}$, where the concentration is highest, and that they are entrained in the turbulent flow, within which the *loitering* effect dominates.

Far below $z_{\mathbf{ref}}$, after sufficient diffusion, dilution, and (for ice) growth, *loitering* becomes negligible and it is assumed that $v_s$ *sweeps* towards the terminal velocity $v_{\mathbf{ter}}$.

Therefore, following the discussions presented in Appendix C we write the governing equation for the *low-order model*, a Burgers-type partial differential equation, together with the associated initial and asymptotic boundary conditions, as:

$$\frac{\partial v_s}{\partial t} + v_s \frac{\partial v_s}{\partial z} = \nu_{\mathbf{t,ef}} \frac{\partial^2 v_s}{\partial z^2},$$

$$v_s(z_{\mathbf{ref}}, t=0) = 0, \qquad v_s(z \ll z_{ref}, t=0) = v_{\mathbf{ter}}, \qquad \lim_{t \to \infty} v_s(z,t) = v_{\mathbf{ter}}. \tag{8}$$

We highlight that, by deliberately encoding the body force within the boundary conditions, the model circumvents the uncertainties and complexities associated with explicitly coupling the body force to the fluid phase and concentration field.

The governing equation requires a closure model to define the threshold distance beyond which turbulent mixing,
characterized by the eddy viscosity $\nu_{\mathbf{t,ef}}$, effectively smooths velocity deviations over a vertical extent denoted by $z_{\mathbf{relax}}$.

To characterize $z_{\mathbf{relax}}$, we can either consider a local or global diffusion–advection balance. In *local diffusion–advection balance*, $z_{\mathbf{relax}}$ is estimated by equating vertical advection and turbulent mixing rates: $v_{\mathbf{ter}} \cdot (\Delta v_s/\delta) \sim \nu_{\mathbf{t,ef}} \cdot (\Delta v_s/\delta^2)$, which yields $\delta \equiv z_{\mathbf{relax}} \sim \nu_{\mathbf{t,ef}}/v_{\mathbf{ter}}$. However, in a *global diffusion–advection balance*, particles sample
turbulent structures of size $L_e$ as they settle; for long-term contrails $L_e$ represents the characteristic size of dominant turbulent structures, typically $10$–$10^3$ m. The advection time across one eddy is defined as $\tau_{\mathrm{adv}} := L_e/v_{\mathbf{ter}}$, during which turbulent diffusion spreads the velocity deviations over a distance $\delta \equiv z_{\mathbf{relax}} \sim \sqrt{\nu_{\mathbf{t,ef}} \cdot \tau_{\mathrm{adv}}} = \sqrt{\nu_{\mathbf{t,ef}} \cdot L_e/v_{\mathbf{ter}}}$. Therefore, we rewrite Eq. (8) (adapted to both the axis direction and the settling sign on defining $\hat{v}_s := |v_s|$), together with the initial condition, as:

$$\frac{\partial \hat{v}_s}{\partial t} + \hat{v}_s \frac{\partial \hat{v}_s}{\partial z} = \nu_{\mathbf{t,ef}} \frac{\partial^2 \hat{v}_s}{\partial z^2},$$

$$\hat{v}_s(z,0) = \begin{cases} v_{\mathbf{ter}}, & z \leq z_{\mathbf{relax}}, \\ 0, & z > z_{\mathbf{relax}}. \end{cases} \tag{9}$$

On using the classic Cole–Hopf transformation, the final closed-form solution is obtained as (see Appendix D):

$$\hat{v}_s(z,t) = v_{\mathbf{ter}} \frac{q_0(z,t)}{q_0(z,t) + q_1(z,t)} \tag{10}$$

where:

$$q_0(z,t) = \mathrm{erfc}\left(\frac{\Delta_0 - v_{\mathbf{ter}}t}{2\sqrt{\nu_{\mathbf{t,ef}}t}}\right) \exp\left(\frac{v_{\mathbf{ter}}^2 t}{4\nu_{\mathbf{t,ef}}} - \frac{v_{\mathbf{ter}}\Delta_0}{2\nu_{\mathbf{t,ef}}}\right), \quad \Delta_0 = z - z_{\mathbf{relax}},$$

$$q_1(z,t) = \mathrm{erfc}\left(-\frac{\Delta_0}{2\sqrt{\nu_{\mathbf{t,ef}}t}}\right). \tag{11}$$





### 4.3 Terminal Velocity, $v_{\text{ter}}$

We present the method used to compute the terminal velocity $v_{ter}$ of randomly oriented spheroidal ice particles in a quiescent fluid under gravity, using the drag model by Ganser (Ganser, 1993). Recall that for spheroids $\mathcal{V} = \frac{4}{3}\pi a^3 \phi$, where $a$ is the equatorial semi-axis and $\phi = c/a$ the aspect ratio. The Ganser model requires the following geometrical definitions: the volume-equivalent diameter $d_v = \left(6\mathcal{V}/\pi\right)^{1/3} = a\left(8\phi\right)^{1/3}$, and sphericity $\Psi = \pi d_v^2/S$ with $S$ being the total surface area of the spheroid, computed as 1) Oblate ($\phi < 1$, $e = \sqrt{1-\phi^2}$): $S = 2\pi a^2\left[1 + \frac{1-e^2}{e}\tanh^{-1}(e)\right]$, 2) Prolate ($\phi > 1$, $e = \sqrt{1-\phi^{-2}}$): $S = 2\pi a^2\left[1 + \frac{\phi}{e}\sin^{-1}(e)\right]$, and 3) Sphere ($\phi = 1$): $S = 4\pi a^2$.

In addition, Ganser defines $d_n$ as the diameter of a sphere whose projected area normal to the motion equals that of the spheroid: thus one writes $A_{\text{proj}} = \pi d_n^2/4$, and since the orientation-averaged projected area satisfies $\langle A_{\text{proj}}\rangle = S/4$ by Cauchy's theorem, one has $d_n = \sqrt{4\langle A_{\text{proj}}\rangle/\pi} = d_s$, where $d_s$ satisfies $S = \pi d_s^2$ ($\Rightarrow d_s = \sqrt{S/\pi}$).

Stokes' shape factor (unbounded fluid) is $K_1 = \frac{1}{3}(d_n/d_v) + \frac{2}{3}(d_s/d_v)$, and the generalized Reynolds number is $Re^* = Re\, K_1\, K_2$ with $Re = \frac{\rho_f v_{\text{ter}} d_v}{\mu_{\text{ef}}}$, where $\mu_{\text{ef}}$ is the effective dynamic viscosity.

Newton's shape factor is fitted as $K_2 = 10^{1.8148(-\log_{10}\Psi)^{0.5743}}$.

With these, Ganser's drag model is given by:

$$C_D = \frac{24}{Re^*}\left(1 + 0.1118\,(Re^*)^{0.6567}\right)K_2 + \frac{0.4305\,K_2}{1 + 3305/Re^*}. \tag{12}$$

To determine the terminal velocity, we write:

$$\tfrac{1}{2}\rho_f v_{ter}^2 C_D\langle A_{\text{proj}}\rangle = (\rho_p - \rho_f)\mathcal{V}g. \tag{13}$$

Let $B = \frac{4(\rho_p - \rho_f)\rho_f g d_v^3}{3\mu_{\text{ef}}^2}$, where $\mu_{\text{ef}} = \frac{\mu}{C(r_{\text{eff}})}$ with $C(r_{\text{eff}})$ being the Cunningham correction factor accounting for slip effects at very small particle sizes (equivalent to non-continuum effects at high Knudsen numbers).

The Cunningham correction factor $C(a^*)$ is defined as:

$$C(r_{\text{eff}}) = 1 + \frac{\gamma}{r_{\text{eff}}}\left(1.257 + 0.400\,e^{-1.1\,\frac{r_{\text{eff}}}{\gamma}}\right). \tag{14}$$

where $\gamma$ is the mean free path and $r_{\text{eff}}$ is the particle's effective radius defined as $r_{\text{eff}} = a\,\phi^{1/3}$.

Substituting $v_{\text{ter}} = \frac{Re\,\mu_{\text{ef}}}{\rho_f d_v}$, gives the following implicit equation:

$$Re^2\,C_D(Re^*) = B. \tag{15}$$

which is solved numerically for $Re$. Therefore, the terminal velocity is retrieved as:

$$v_{\text{ter}} = \frac{Re\,\mu_{ef}}{\rho_f d_v}. \tag{16}$$

Notably, because $\langle A_{\text{proj}}\rangle$ is the orientation-averaged projected area, Ganser's drag model inherently accounts for random tumbling orientation.





### 4.4 Preliminary Results for the Bulk Settling Velocity $v_s$ and Comparison

To elucidate the bulk-settling closed-form formula, we evaluate Eq. (10) by fixing the equatorial radius at $25\,\mu m$
while varying the particle shape index $\phi$ (note that here, shape index $\phi$ and equatorial radius $a$ are fixed, resulting
in fixed $v_{\mathbf{ter}}$ which is not a realistic scenario and hence, Fig. 2 only shows the behavior of the closed-form solution
for the bulk settling velocity). As shown in Fig. 2, smaller ice crystals remain suspended for longer durations due
to the dominance of turbulent mixing (*loitering*) over gravitational settling (*preferential sweeping*). In contrast, for
larger particles, gravitational effects rapidly outweigh turbulent dispersion, resulting in the bulk settling velocity $v_s$
converging more quickly to the individual terminal velocity $v_{\mathbf{ter}}$.

To evaluate the performance of the low-order model against reported reductions in bulk settling speed attributed
to the loitering effect, we found that a direct comparison is not straightforward, primarily because our model is
formulated within a different framework. However, by employing the turbulent quantities defined in the bulk settling
velocity $v_s(z,t)$, it is possible to replicate the experimental setup reported in (Nielsen, 1993). Specifically, we define the
bulk-averaged settling velocity as $\bar{v}_s = \langle c_N, v_s \rangle / \langle c_N \rangle$, and similarly $\bar{v}_{\mathbf{ter}} = \langle c_N, v_{\mathbf{ter}} \rangle / \langle c_N \rangle$. The turbulence intensity
ratio is then introduced as $s := \sigma / \bar{v}_{\mathbf{ter}}$, where, upon substituting the turbulent quantities into the $v_s$ formulation, we
set $\sigma := \nu_{\mathbf{t,ef}} / L_\sigma$. Thus, the intensity ratio becomes $s = \nu_{\mathbf{t,ef}} / (L_\sigma \bar{v}_{\mathbf{ter}})$. In our experiments, the turbulence intensity
is varied by adjusting $\nu_{\mathbf{t,ef}}$, while $L_\sigma$ is fixed to ensure that $s$ spans the range $0.1 \lesssim s \lesssim 20$, thereby covering the
$x$-axis of the random-walk model in (Nielsen, 1993). Furthermore, the persistence of coherent eddy structures is
quantified by the index $A_e := \sigma \tau_{\mathrm{adv}} / L_e = s L_e / L_\sigma$. For the present comparison, we fix $A_e = 1$ and directly compare
our settling velocity reduction with the corresponding results reported for $A_e = 1$ in (Nielsen, 1993).

Therefore, we write $L_e = L_\sigma / s$, which provides a progressive refinement of $L_e$. We then solve the full contrail system
of equations for multiple peaks of the initial ice supersaturation profile, $s_{i,\mathrm{peak}}(z,0)$, and compute the corresponding
settling ratio as: $\frac{\bar{v}_s}{\bar{v}_{\mathbf{ter}}}\left(\frac{\sigma}{\bar{v}_{\mathbf{ter}}}\right) := \frac{1}{T}\int_0^T \frac{\langle v_s, c_N \rangle}{\langle v_{\mathbf{ter}}, c_N \rangle}\,\mathrm{d}t$. The numerical solution is obtained following the procedures outlined
in Sec. 5. The comparison is presented in Fig. 3, which demonstrates generally good agreement with the random-walk
model in both the functional behavior and the overall range. In particular, a close match is observed for cases of lower
turbulence intensity or larger terminal velocities, the latter being characteristic of larger particles. The deviations
observed in the present Eulerian bulk-settling model are likely attributable to differences in the underlying turbulent
structure and the modeling framework.

### 5 Separability of the System of Equations: Minimal Representation of the 3D Framework

In this study we solve the minimal representation of the three-dimensional framework and examine differences
in plume properties predicted by the classical spherical-particle model versus the proposed habit–dynamic model.
Specifically, although the governing equations (Eqs. 3) are inherently three-dimensional, requiring a fully resolved
3D mesh and often forcing a trade-off between resolution and domain size, the system admits a separable structure




**Figure 2.** Bulk settling velocity ratio $\frac{v_s(z,t)}{v_{\mathrm{ter}}}$ at fixed equatorial radius $a = 25\,\mu\mathrm{m}$ and varying shape index $\phi$.

under physically justified assumptions. Exploiting this separability permits high-resolution solutions over extended domains without resorting to coarse-grained approximations and high computational times.

We initialize the contrail by releasing ice particles into a three-dimensional region immediately following the earlier stages that the contrail has been through (roughly, after 10 to 20 minutes of contrails birth), prescribing initial fields as:

$$c_N(\mathbf{x}, 0) = c_{N,0}, \quad m(\mathbf{x}, 0) = m_0, \quad \phi(\mathbf{x}, 0) = \phi_0. \tag{17}$$

The volumetric growth rate $f_{\mathcal{V}}$ depends on ambient temperature $T(x, y, z, t)$, but observational datasets typically have horizontal resolution of order kilometers and exhibit weak horizontal gradients. Accordingly we assume $T = T(z, t)$, and, $s_i = s_i(z, t)$.





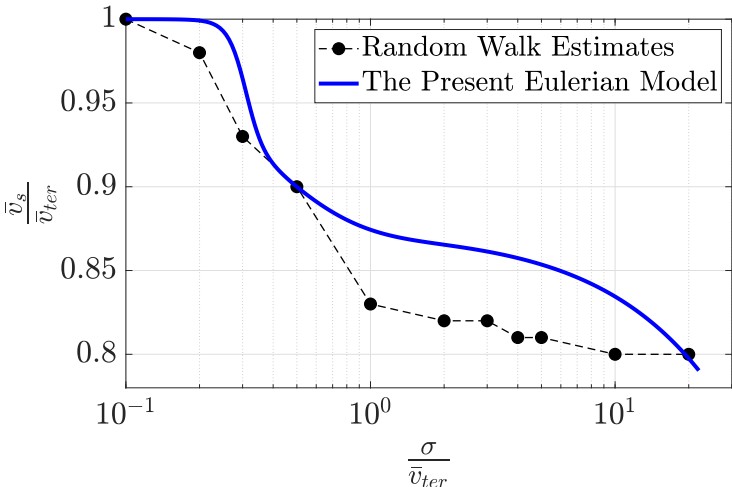

**Figure 3.** Settling velocity ratio $\frac{\bar{v}_s}{\bar{v}_{\text{ter}}}$ as a function of turbulent intensity $\frac{\sigma}{\bar{v}_{\text{ter}}}$. Blue line: The present Eulerian model, black dots: random walk estimates (Nielsen, 1993).

**Remark.** Suppose a set-up when $f_{\mathcal{V}} = 0$. Therefore, each particle's mass remains fixed and the diffusive operators in the $m$-equation do not introduce any net mass transport or source. Instead, they capture the turbulent mixing of particles carrying different masses between adjacent control volumes, acting solely to homogenize the *mean* mass field by damping spatial gradients. Consequently, if $m(\mathbf{x}, 0) = m_0$ is uniform (so that $\nabla m = 0$), no smoothing occurs and the particle ensemble remains constant in the absence of growth. In other words, the diffusive contributions in the $m$-equation are strictly *inward* in effect: they act to smooth out spatial gradients in the mean particle mass field, thereby reducing heterogeneities introduced by nonuniform growth or initial conditions, rather than inducing any outward flux of mass. Therefore, with $T = T(z,t)$, and $s_i = s_i(z,t)$, we have $f_{\mathcal{V}} = f_{\mathcal{V}}(z,t)$ and together with uniform mass distribution in the horizontal plane, we can write $m = m(z,t)$.

Therefore, we adopt the following separable ansatz:

$$c_N(x, y, z, t) = F(x, y, t)\, g(z, t), \tag{18}$$

with a closure for the turbulent diffusivity tensor:

$$\widetilde{\mathcal{D}}_{ij}(c_N) \longrightarrow \begin{cases} \widetilde{\mathcal{D}}_{xx}\big(F(x, y, t)\big), \\ \widetilde{\mathcal{D}}_{yy}\big(F(x, y, t)\big), \\ \widetilde{\mathcal{D}}_{zz}\big(g(z, t)\big). \end{cases} \tag{19}$$

Enforcing vanishing fluxes at $x, y \to \pm\infty$ and $z \to \pm\infty$ together with the normalization $\int\limits_{-\infty}^{+\infty} g(z,t)\,\mathrm{d}z = 1$, and the system decouples into:



$$\frac{\partial F}{\partial t} + w_x(x,y,t)\frac{\partial F}{\partial x} + w_y(x,y,t)\frac{\partial F}{\partial y} = \frac{\partial}{\partial x}\Big(\widetilde{\mathcal{D}}_{xx}(F)\frac{\partial F}{\partial x}\Big) + \frac{\partial}{\partial y}\Big(\widetilde{\mathcal{D}}_{yy}(F)\frac{\partial F}{\partial y}\Big),$$

$$\frac{\partial g}{\partial t} + v_s(z,t)\frac{\partial g}{\partial z} = \frac{\partial}{\partial z}\Big(\widetilde{\mathcal{D}}_{zz}(g)\frac{\partial g}{\partial z}\Big) - g\frac{\partial v_s}{\partial z},$$

$$\frac{\partial m}{\partial t} + v_s(z,t)\frac{\partial m}{\partial z} = \frac{\partial}{\partial z}\Big(\widetilde{\mathcal{D}}_{zz}(g)\frac{\partial m}{\partial z}\Big) + \frac{2\widetilde{\mathcal{D}}_{zz}(g)}{g}\frac{\partial m}{\partial z}\frac{\partial g}{\partial z} + \rho_{\text{dep}}\, f_{\mathcal{V}}(z,t),$$

$$\frac{\partial \phi}{\partial t} + v_s(z,t)\frac{\partial \phi}{\partial z} = \frac{\Gamma^*\big(T(z,t),s_i(z,t)\big)-1}{\Gamma^*\big(T(z,t),s_i(z,t)\big)+2}\frac{\phi}{\mathcal{V}}\, f_{\mathcal{V}}(z,t),$$

$$|v_s(z,t)| = v_{\mathbf{ter}}\frac{q_0(z,t)}{q_0(z,t)+q_1(z,t)},$$

$$q_0(z,t) = \text{erfc}(\frac{\Delta_0 - v_{\mathbf{ter}}t}{2\sqrt{\nu_{\mathbf{t,ef}}t}})\exp\Big(\frac{v_{\mathbf{ter}}^2 t}{4\nu_{\mathbf{t,ef}}} - \frac{v_{\mathbf{ter}}\Delta_0}{2\nu_{\mathbf{t,ef}}}\Big), \quad \Delta_0 = z - z_{\mathbf{relax}},$$

$$q_1(z,t) = \text{erfc}(-\frac{\Delta_0}{2\sqrt{\nu_{\mathbf{t,ef}}t}}).$$

$$\mathrm{d}X = -\frac{X-\mu}{\tau}\,\mathrm{d}t + \sigma_X\,\mathrm{d}W_t, \quad X \in \{\tilde{\mathcal{D}}_{xx}, \tilde{\mathcal{D}}_{yy}, \tilde{\mathcal{D}}_{zz}\},$$

$$\mathcal{V}(z,t) = \frac{m(z,t)}{\rho_{dep}}. \tag{20}$$

Equations (20) govern the fully coupled yet separable evolution of number concentration, mass (with growth), shape, with stochastic diffusivity and bulk settling-velocity.

The initial plume, representing the neacliated soot particles after a short time of aircraft travel, $T$, can be defined in multiple ways, for example the following Gussian plume:

$$c_N(x,y,z,0) = F(x,y,0)\,g(z,0) = \frac{N_{tot}}{2\pi\sigma_x\sigma_y}e^{-\left(\frac{(x-x_0)^2}{2\sigma_x^2} + \frac{(y-y_0)^2}{2\sigma_y^2}\right)}\frac{1}{\sigma_z\sqrt{2\pi}}e^{-\left(\frac{(z-z_0)^2}{2\sigma_z^2}\right)}. \tag{21}$$

giving $\iiint c_N(x,y,z,0)\,dx\,dy\,dz = N_{\text{tot}}$ where $N_{\text{tot}} = s\,\text{EI}_N\,\dot{m}_f\,T$. Here, $s$ is the activation/survival factor (the fraction of emitted soot particles that actually form ice crystals), $\text{EI}_N$ denotes the number emission index (particles emitted per kilogram of fuel) and $\dot{m}_f$ is the engine fuel mass flow rate ($\text{kg}\,\text{s}^{-1}$), and $T = \frac{L}{v_{\text{ac}}}$ is the total time the aircraft travels across the domain, with $L$ being the traveled distance and $v_{\text{ac}}$ the aircraft speed.

In this research, we normalize both $g(z,t)$ and $F(x,y,t)$ by writing: $\tilde{F} \equiv \frac{2\pi\sigma_x\sigma_y}{N_{tot}}F(x,y,t)$, and $\tilde{g} \equiv \sigma_z\sqrt{2\pi}g(z,t)$. Furthermore, we define ice-water content as $\text{IWC}(x,y,z,t) := g(z,t)F(x,y,t)m(z,t)$. We use full width at half maximum to define $L \equiv 2\sigma_x\sqrt{2\ln 2}$. Therefore, we approximate $\text{IWC}(x,y,z,t) = \frac{2\,s\,\text{EI}_N\,\dot{m}_f\,\sqrt{2\ln 2}}{(2\pi)^{\frac{3}{2}}\sigma_y\sigma_z v_{\text{ac}}}\tilde{g}(z,t)\tilde{F}(x,y,t)m(z,t)$.

For initialization in our synthetic simulations, the particle emission rate (particles per second) is defined as $\text{E} := \text{EI}_N\,\dot{m}_f$. We define the equivalent linear number density along the flight track (particles per meter) as $\text{N}_{\text{lin}} := \frac{\text{E}}{v_{\text{ac}}}$. Therefore, we write:

$$\text{IWC}(x,y,z,t) = \frac{2\,s\,\text{N}_{\text{lin}}\,\sqrt{2\ln 2}}{(2\pi)^{3/2}\sigma_y\sigma_z}\,\tilde{g}(z,t)\,\tilde{F}(x,y,t)\,m(z,t). \tag{22}$$



We set $\mathrm{N_{lin}} = \frac{\mathrm{EI}_N \dot{m}_f}{v_{ac}} \approx \frac{4\times10^{14}\times1.3}{240} \approx 2.17 \times 10^{12}\,\mathrm{m^{-1}}$ (Li et al., 2023), which is close to the value reported in (Unterstrasser and Gierens, 2010b). The survival/activation fraction is set to $s = 0.35$ (Unterstrasser and Gierens, 2010b). In addition, we estimate $\sigma_z$, $\sigma_y$ and $\sigma_x$ by solving the diffusion equation for a moving point source (representing continuous emission over an aircraft trajectory of length $L \approx 50$ km, inside an ice-supersaturated region (ISSR) at an

airspeed of $v_{\mathrm{ac}} = 240\,\mathrm{m\,s^{-1}}$), and equating the resulting diffused cross-sectional area to that of a Gaussian plume.

Notably, because this research does not involve simulation or parameterization of the earlier contrail stages, the timing in our simulations refers to the period after those initial stages. Those early stages typically last several minutes (up to 10–20 minutes) before the jet-induced and wake velocities decay and the advective velocity switches to the background wind.

Moreover, we note that the relative humidity in the expanding contrail core remains close to ice saturation because entrainment of fresh, unsaturated air occurs on a much longer timescale (hours) than crystal growth (minutes). To represent this effect in our synthetic simulations we employ an auxiliary diffusion–reaction equation: $\frac{\partial s_i}{\partial t} = \kappa \frac{\partial^2 s_i}{\partial z^2} - \alpha\, g(z,t)\, m(z,t)$, where $\kappa$ parameterizes diffusion and $\alpha$ represents the balance between supersaturation depletion by ice growth and its recovery. The initial ice-supersaturation layer in this study is taken to be approximately

1.3 km thick; however, supersaturation within the layer is not uniform; it follows a Gaussian profile that peaks near flight level (the reference altitude), decays smoothly to near zero above the layer, and decreases below the layer to about $-0.08$.

## 5.1 Solution Methodology

In this work, we employ the directional-ODE discretization framework, as introduced in (Jafarimoghaddam et al.,

2025). This method recasts the discretization of partial differential equations, particularly advection-diffusion equations (ADEs), into representative ODEs along either the spatial or temporal dimension (see (Jafarimoghaddam et al., 2025) for comprehensive details). Notably, the directional-ODE approach has been shown to offer advantages over conventional implicit schemes in terms of both accuracy and computational efficiency. In our implementation, we adopt a first-order predictor-corrector algorithm, following the formulation presented in (Jafarimoghaddam et al.,

470    2025).

## 6 Contrail Plume Simulation Results

## 6.1 Impact of Diffusion Blocking Coefficient on Horizontal Contrail Evolution

In this section, we investigate the influence of the nonlinear diffusion blocking coefficient $\beta$ on the evolution of contrail plumes. A synthetic wind field is generated using the composite inviscid wind model coupled with a stochastic

atmospheric turbulence structure. Three distinct contrail initializations are considered, each modeled as a narrow Gaussian plume located at different positions in the domain. Two diffusion scenarios are analyzed: (1) no diffusion





blocking, i.e., $\beta_x = \beta_y = 0$, and (2) moderate diffusion blocking, with $\beta_x = \beta_y = 1000$. The mean wind magnitudes are normalized to permit high grid resolution in a limited computational domain and to focus exclusively on the role of the diffusion-blocking coefficient. Simulations are conducted over a period of 5 hours. As illustrated in Fig.

4, the introduction of the nonlinear blocking term ($\beta = 1000$) substantially weakens the spatial dispersion of the contrails, resulting in more confined plume structures, mainly following the characteristic lines. In contrast, the zero-blocking case ($\beta = 0$) leads to significantly broader and more diluted contrail fields. These results confirm that the nonlinear diffusion term plays a critical role in limiting horizontal spreading under certain conditions. However, the calibration of $\beta$ remains an open question. Improved constraint of this parameter requires systematic comparison

with observational data, particularly from ground-based contrail imagery under diverse meteorological conditions, to ensure realistic parameterization of the diffusion blocking mechanism.

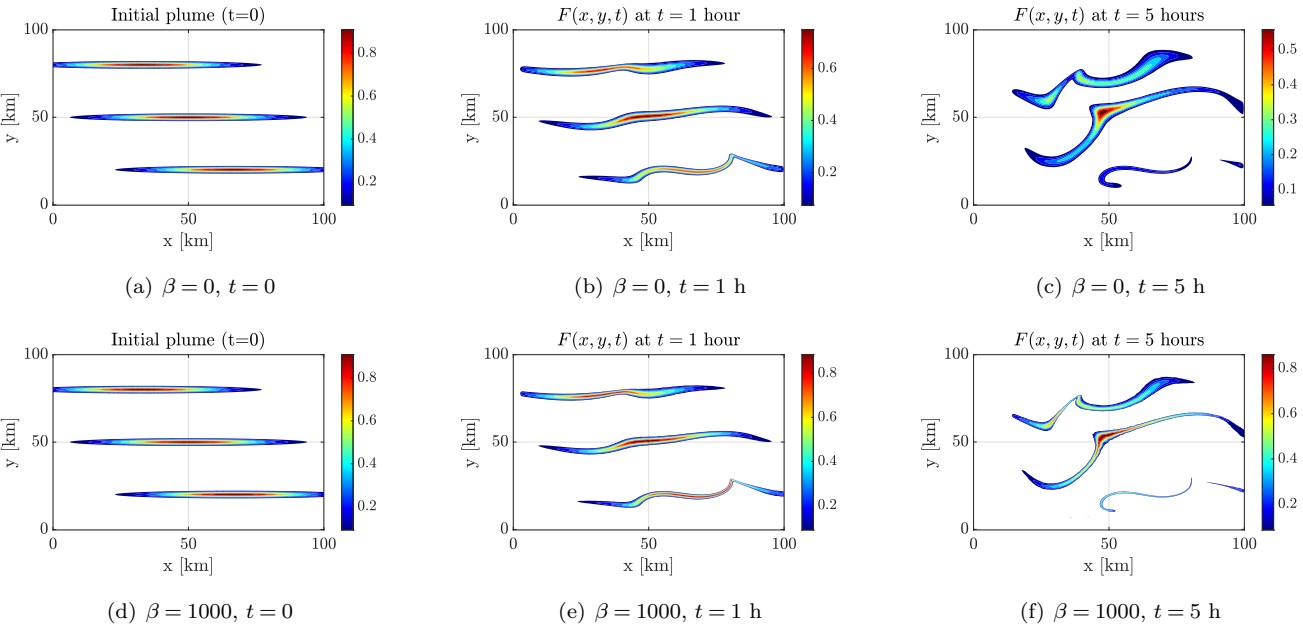

**Figure 4.** Horizontal plume simulation for $F(x,y,t)$ under different diffusion blocking coefficients. Top row: $\beta = 0$; bottom row: $\beta = 1000$.

## 6.2    Qualitative Comparison of Crystal Habits

Some *in situ* and remote-sensing observations indicate that persistent contrails (and contrail cirrus) tend to contain a higher proportion of plate-like ice crystals, with columnar or needle-shaped crystals appearing only transiently

(often at the cloud edges) and falling out or sublimating more rapidly. Most of the available reports pertain to young contrails, typically those persisting for less than one hour. For example, image data often show plate-like ice crystal habits, and it is hypothesized that a lack of water vapour budget prevents the crystals from growing into more





complex, rosette-like forms (Jones et al., 2012; Heymsfield et al., 1998; Schumann et al., 2017). In addition, *in situ* samples of contrail ice crystals taken at $-61\,^{\circ}$C revealed several ice habits: hexagonal plates (75%), columns (20%), and a few triangular plates (about 5%) (Yang et al., 2010). In addition, (Sussmann, 1997) reports on the contrails' crystal shape can consist of regular habits dominated by hexagonal plates.

Notably, several factors influence the transient habit dynamics which need to be considered in experimental setups. The most important of these are the ice crystal's age/history, local ice supersaturation, and temperature, all of which feed back into crystal growth and the bulk settling speed. In turn, the bulk settling speed determines the duration for which an ice crystal is exposed to a particular level of ice supersaturation and temperature. In this respect, the literature appears to lack fully-controlled experimental setups to document the cycling behavior of habits' sedimentation dynamics.

Nonetheless, in this study we found that our results are qualitatively comparable with existing observational data in (Yang et al., 2010). In particular, our simulations indicate that the dominant crystal habits are plate-like ice crystals, with columnar crystals gradually migrating toward the edges of the contrail layer over time. Although higher ice supersaturation typically provides more favorable conditions for the formation of needle-like, bullet rosette and columnar crystals, the coupling between settling velocity and crystal growth suggests that these crystal types are advected out of the ice-supersaturated layer and begin to sublimate sooner than plate-like crystals. Specifically, we conducted a simulation at $-61\,^{\circ}$C (similar to the observations reported in (Yang et al., 2010)) at the reference altitude with an initial layer of ice supersaturation peaking locally at up to 17%, i.e., $s_{i,\text{peak}}(z,0) \approx 17\%$. Our results show that plate-like crystals rapidly become dominant, exceeding 80% within about two hours, before gradually decreasing (see Fig. 5-**a**, together with Fig. 5-**b**, which presents the mean size distribution of habits where a number concentration threshold $\tilde{g}(z,t) > 10^{-3}$ is applied). At the same reference temperature, but with a layer of ice supersaturation peaking locally at up to 27%, i.e., $s_{i,\text{peak}}(z,0) \approx 27\%$, we derived the time evolution of the habit percentages (see Fig. 6). In this case, our results demonstrate that during the first hour, columnar crystals dominate. Subsequently, plate-like crystals become the dominant habit before gradually decreasing, while columnar crystals are advected toward the edges of the contrail layer.

### 6.3  Vertical Evolution of Contrail Plume

In this section, we present simulation results for an initial Gaussian plume in 3D. The initial plume follows the Gaussian distribution given in Sec. 5. At the reference altitude $z_{\text{ref}} = 0$, the temperature is $T = 212.15\,\text{K}$, and the initial ice supersaturation $s_i$ peaks at about 17%. The ice crystals are initially spherical with a radius of $1\,\mu\text{m}$, i.e., $\phi(z,0) = 1$. The number of vertical grids is varied until suitable numerical convergence is achieved. In this respect, the $z$-domain from 1000 to $-2500$ meters is discretized into 2000 grids (giving a resolution of 1.75 meters), at which point we found that the solutions converge.





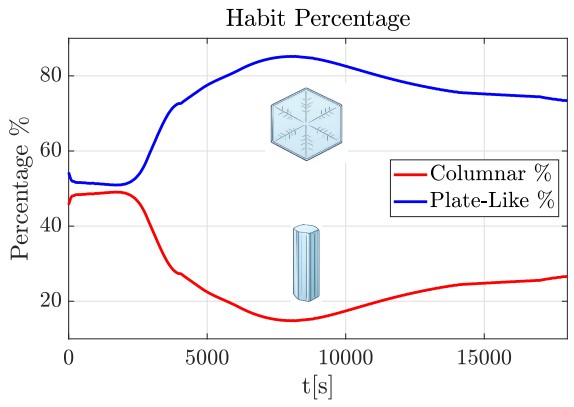

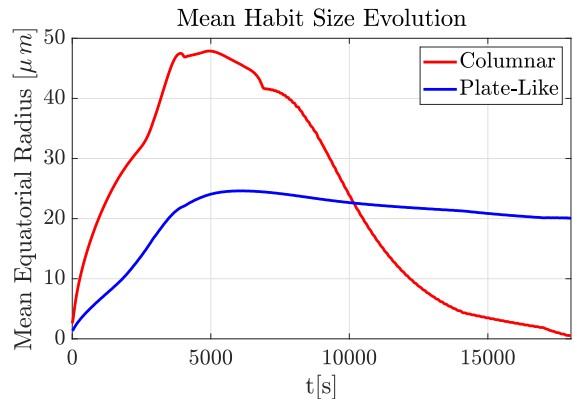

(a) Crystal shape distribution at different times

(b) Crystal size distribution at different times

**Figure 5.** Crystal shape and size distributions as functions of time at the reference temperature $T = -61\,°\mathrm{C} = 212.15\,\mathrm{K}$ and $s_{i,\mathrm{peak}}(z,0) \approx 17\%$.

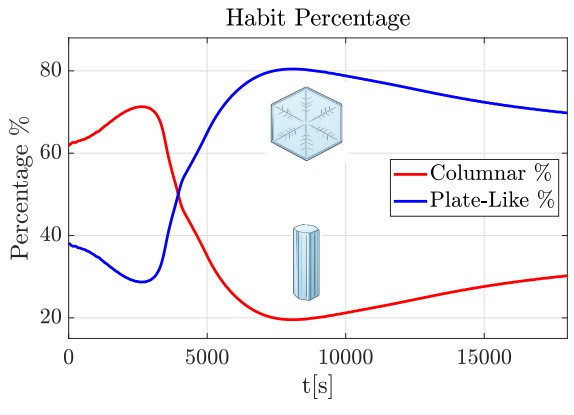

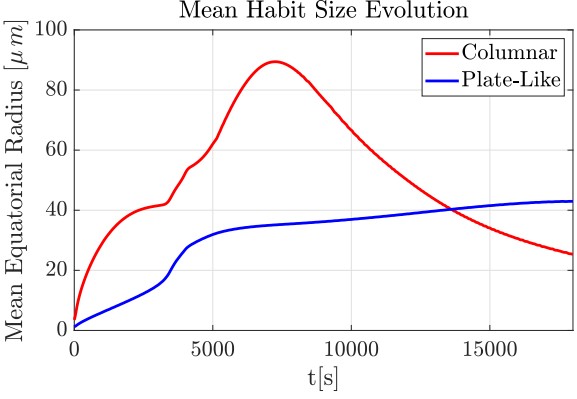

(a) Crystal shape distribution at different times

(b) Crystal size distribution at different times

**Figure 6.** Crystal shape and mean size distributions as functions of time at the reference temperature $T = -61\,°\mathrm{C} = 212.15\,\mathrm{K}$ and $s_{i,\mathrm{peak}}(z,0) \approx 27\%$.

The simulations are conducted for two distinct scenarios: 1) **Spherical Model**, which assumes $\phi(z,t) = 1, \forall t > 0$, meaning that habit dynamics are decoupled from the number and mass ADEs, and 2) **Habit Model**, which solves the full system of ADEs.

    We present the static contours at different times in the $x$–$z$ plane along the track, i.e., along the center of the plume where maximum concentration is expected. The key quantities of number and mass concentrations are depicted.

Specifically, in Fig. 7, the IWC of the two scenarios is compared and labeled as $\mathrm{IWC}_h$ (for the Habit Model) and





IWC$_s$ (for the Spherical Model). Fig. 8 shows a comparison for number concentration, denoted $c_{N,h}$ for the Habit Model and $c_{N,s}$ for the Spherical Model.

From Fig. 7, we observe that IWC shows different behaviors in the Habit and Spherical Models. In the Habit Model, the plume center naturally remains at the top due to the fallstreak of heavier columnar crystals, which on average also have faster settling velocities, particularly at lower Reynolds numbers. These crystals soon leave the supersaturation region and sublimate, leaving the plume center to persist longer near the plume top. In the Spherical Model, however, the need for an additional updraft correction is more pronounced because it assigns the same settling velocity to equal-mass crystals. Moreover, it does not incorporate the influence of crystal shape on growth mechanisms, such as capacitance.

Following similar reasoning, for the Spherical Model, number concentration remains more continuous. However, in general, number concentration is less prone to strong deviations between the two models. Mathematically, this is because number concentration only accounts for the influence of settling velocity, which itself is implicitly affected by the growth term (see Fig. 8).

We also plot the quantities of interest, namely, the normalized number concentration $\tilde{g}(z,t)$, individual mass $m(z,t)$, ice water content IWC$(z,t)$, and ice crystal shape function $\phi(z,t)$, at different times (see Fig. 9). From these figures, it becomes clear that heavier columnar crystals leave the ice-supersaturation region more quickly while experiencing different IGF factors along the $z$-direction. Over time, the percentage of columnar crystals decreases, leaving the plume dominated by plate-like crystals.

Figure 10-**a** shows a comparison between the equatorial radius distribution in the Habit Model and the radius distribution in the Spherical Model at different times. Figure 10-**b** shows a similar comparison, but now for the effective radius, from which we observe that the effective radius in the Habit Model is larger at earlier times when growth is more dominant, and that the two models gradually converge as sublimation sets in. Figure 10-**c** compares the bulk settling velocity in the Habit and Spherical Models, showing that closer to the reference altitude the deviation is less pronounced due to the *loitering* effect encoded in the bulk settling velocity. However, heavier columnar crystals farther downstream show greater deviations from the Spherical Model.

The time evolution of the quantities, $\tilde{g}(z,t)$, $m(z,t)$, and $r_{\mathbf{eff}}(z,t)$, for both the Spherical and Habit Models are presented as contour plots in Fig. 11, Fig. 12, and Fig. 13 respectively.

Specifically, from Fig. 11 we observe that the normalized number concentration $\tilde{g}(z,t)$ is similar for the two models, except that streamline convergence (mainly due to the term $\tilde{g}\frac{\partial v_s}{\partial z}$) downstream is more pronounced in the Habit Model. From Figs. 12 and 13 we observe that crystals grow more in the Habit Model at earlier times, and the vertical extent of $m(z,t)$ in the Habit Model is smaller than that of the Spherical Model.

Figures 14-**a** and 14-**b** show the time history of the equatorial radius $a(z,t)$ and $\phi(z,t)$ for the Habit Model, from which the dominance of plate-like crystals in the contrail core is evident, consistent with the discussions presented earlier.







(a) IWC for the habit model

(b) IWC for the spherical model

**Figure 7.** Comparison of ice water content (IWC) between the two scenarios: (a) habit model and (b) spherical model.





(a) Number concentration for the habit model

(b) Number concentration for the spherical model

**Figure 8.** Comparison of number concentration ($c_N$) between the two scenarios: (a) habit model ($c_{N,h}$) and (b) spherical model ($c_{N,s}$).



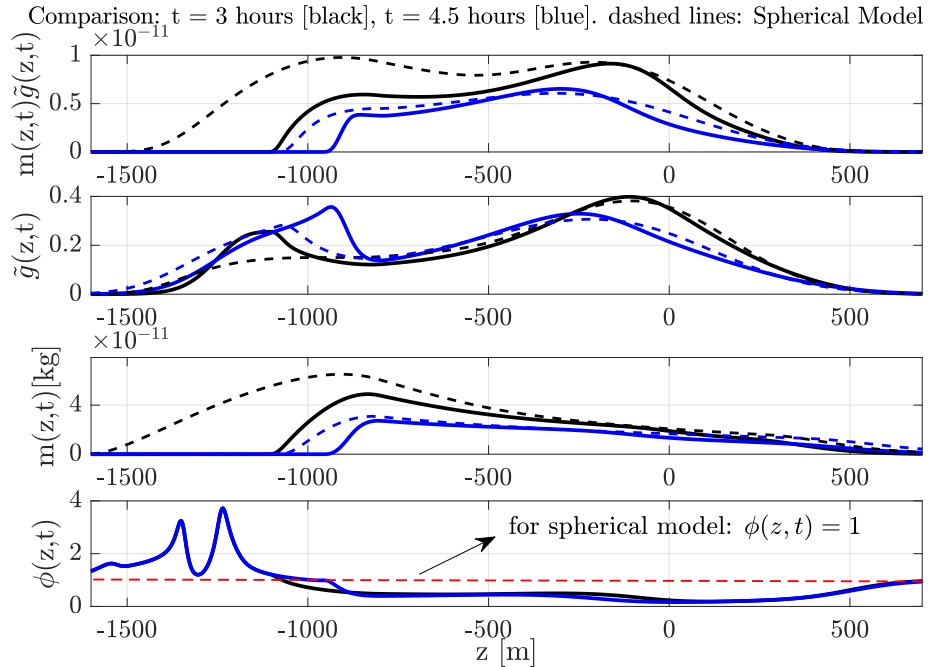

(a) Comparison of $\tilde{g}(z,t), m(z,t), \phi(z,t)$, and $\tilde{g}(z,t)\,m(z,t)$: $t = 0.5$ hours (black) and 2 hours (blue).

(b) Comparison of $\tilde{g}(z,t), m(z,t), \phi(z,t)$, and $\tilde{g}(z,t)\,m(z,t)$: $t = 3$ hours (black) and 4.5 hours (blue).

**Figure 9.** Vertical plume properties: comparison between **Spherical Model** and **Habit Model**.





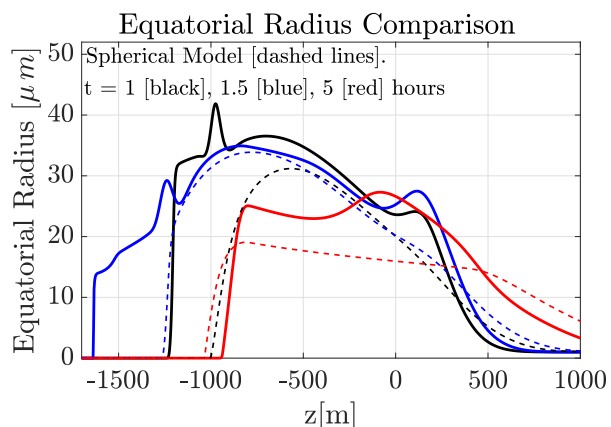

(a) Equatorial radius comparison: $a(z,t)$.

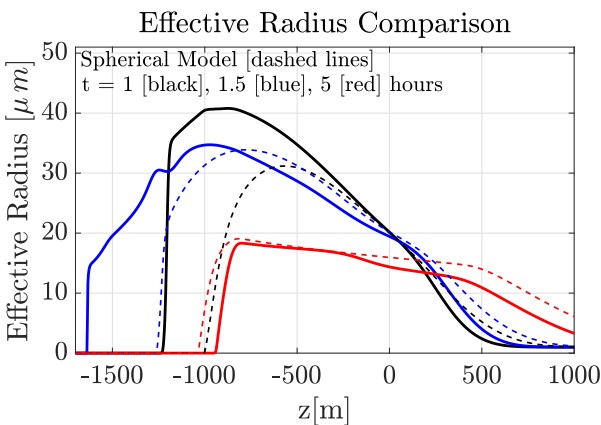

(b) Effective radius comparison: $r_{\mathbf{eff}}(z,t)$.

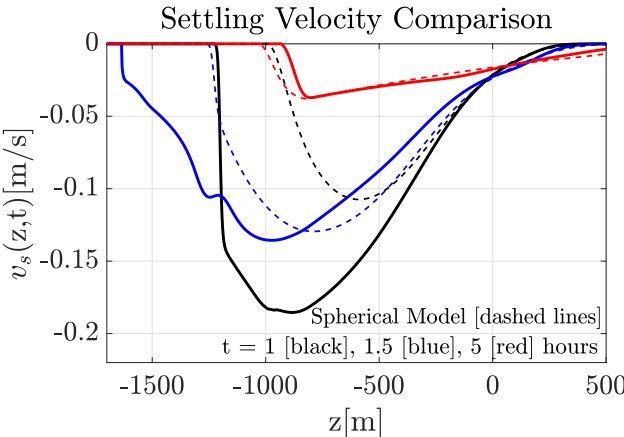

(c) Settling velocity comparison: $v_s(z,t)$.

**Figure 10.** Vertical plume properties: comparison between the **Spherical Model** and **Habit Model** for $a(z,t)$, $r_{\mathbf{eff}}(z,t)$, and $v_s(z,t)$.





Finally, we define a metric to measure the total absolute deviation of IWC of the Habit Model compared to the
Spherical Model. The $L_1$-based deviation metric for scenario $S_i := s_{i,peak}(z,0)$ is:

$$r_{\text{IWC}}(S_i) := \frac{1}{\bar{t}} \int\limits_0^{\bar{t}} \left( 1 + \left| \frac{\overline{\text{IWC}}_h(t)}{\overline{\text{IWC}}_s(t)} - 1 \right| \right) \mathrm{d}t. \tag{23}$$

where:

$$\overline{\text{IWC}}_m(t) := \frac{1}{|\Omega_z|} \int\limits_{\Omega_z} \text{IWC}_m(z,t)\,\mathrm{d}z, \qquad m \in \{h,s\}. \tag{24}$$

Fig. 15 illustrates the above quantity at $\bar{t} \approx 30$ minutes, when the deviation is most pronounced, and at $\bar{t} \approx 5$ hours,
when sublimation dominates and the deviation becomes less significant.

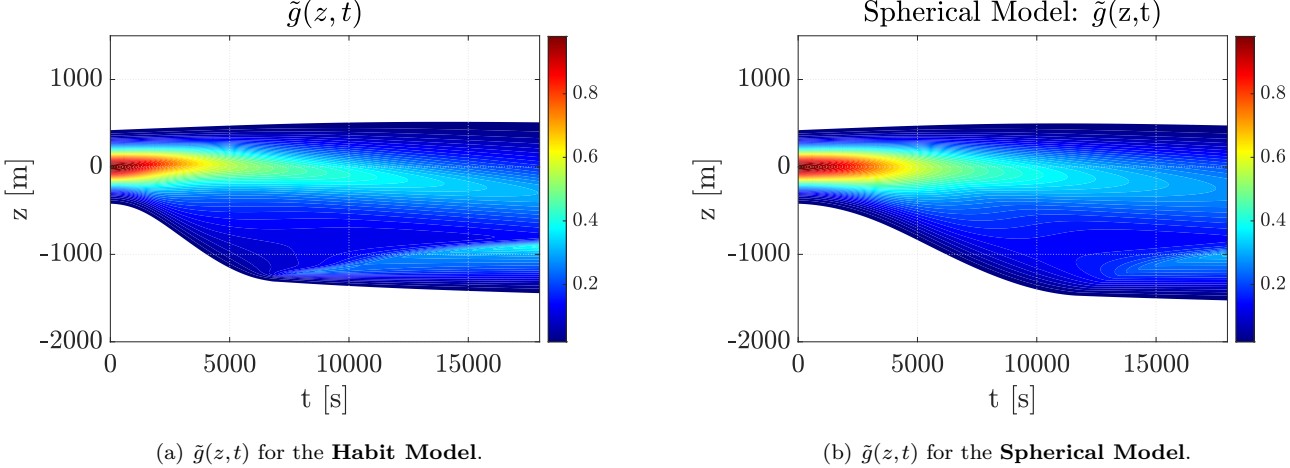

(a) $\tilde{g}(z,t)$ for the **Habit Model**.                    (b) $\tilde{g}(z,t)$ for the **Spherical Model**.

**Figure 11.** Comparison between the **Spherical Model** and **Habit Model** for the normalized number concentration $\tilde{g}(z,t)$.

# 7   Conclusion

We presented an Eulerian, multi-physics framework for long-term contrail evolution that retains two moments of
the population-balance equation (PBE), thereby ensuring conservation of particle number and mass. The model
incorporates spatiotemporally varying, nonlinear diffusion coefficients to represent diffusion-blocking effects, a novel
settling-velocity formulation that accounts for bulk settling in turbulent flows, a tracking-field equation for habit
dynamics coupled to microphysical growth and gravitational settling, and a recently-proposed discretization approach
that improves numerical efficiency and accuracy compared to conventional solvers. Under mild regularity assumptions
the governing system admits a separable structure, which is well suited for large-scale simulations by striking a
favorable balance between accuracy and computational cost.





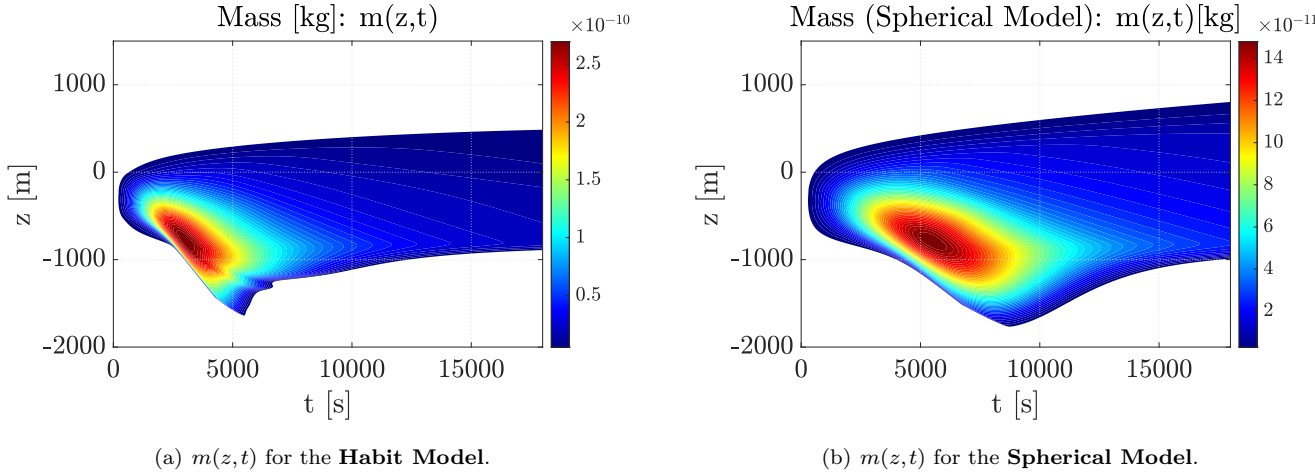

(a) $m(z,t)$ for the **Habit Model**.

(b) $m(z,t)$ for the **Spherical Model**.

**Figure 12.** Comparison between the **Spherical Model** and **Habit Model** for the individual mass field $m(z,t)$.

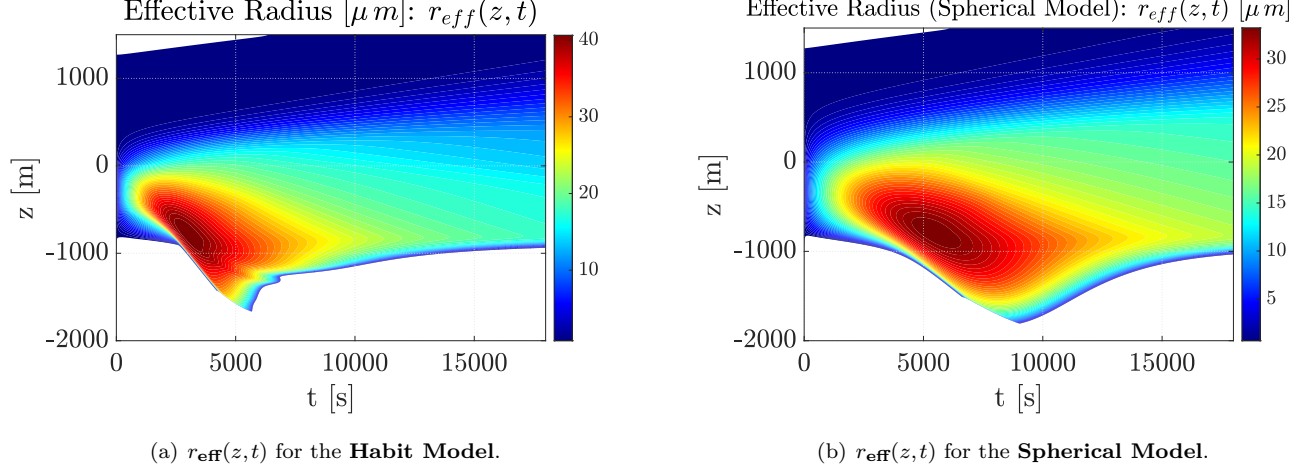

(a) $r_{\mathbf{eff}}(z,t)$ for the **Habit Model**.

(b) $r_{\mathbf{eff}}(z,t)$ for the **Spherical Model**.

**Figure 13.** Comparison between the **Spherical Model** and **Habit Model** for the effective radius $r_{\mathbf{eff}}(z,t)$.





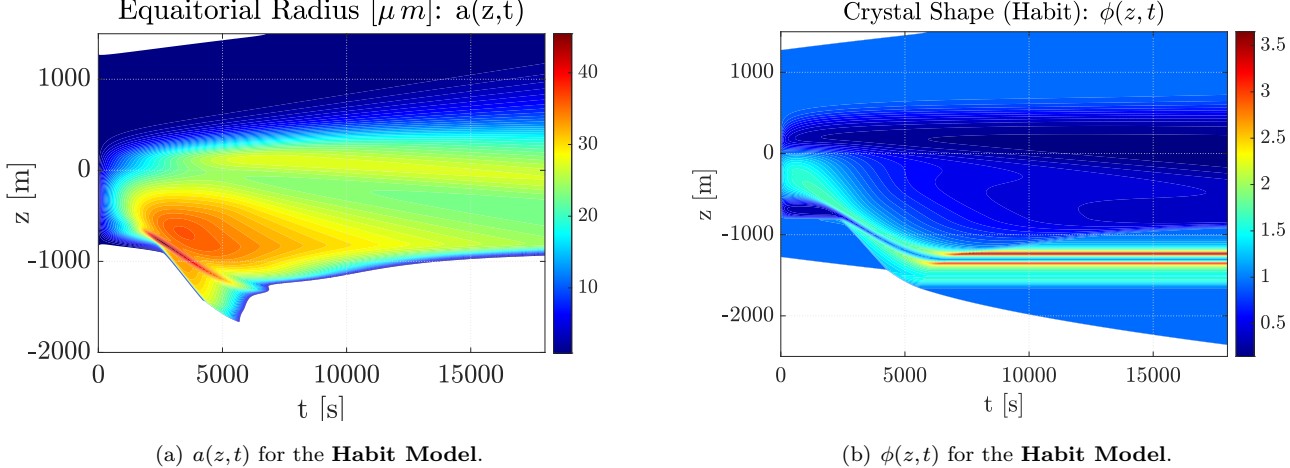

(a) $a(z,t)$ for the **Habit Model**.

(b) $\phi(z,t)$ for the **Habit Model**.

**Figure 14.** Vertical plume properties of the **Habit Model**: equatorial radius $a(z,t)$ (left) and shape index $\phi(z,t)$ (right).

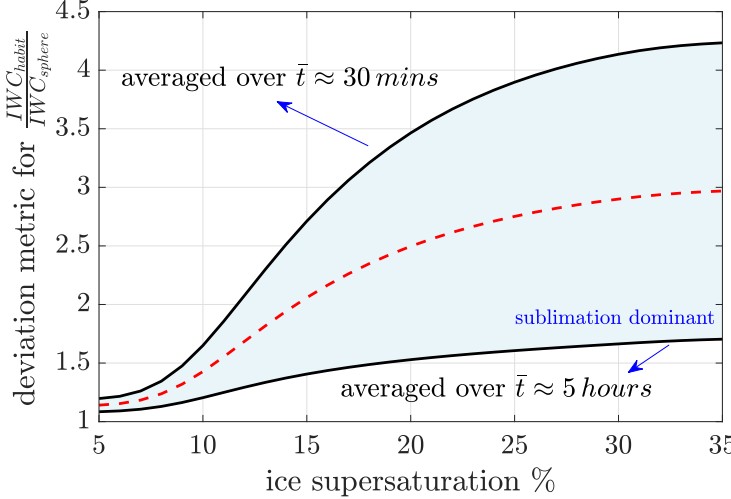

**Figure 15.** Comparison between the **Habit Model** and the **Spherical Model**: illustration of the deviation metric for IWC, $r_{\text{IWC}}(S_i)$.



The framework is readily extensible: at modest additional cost it can be applied to the full vertical plane (e.g., an $x$–$z$ domain to explicitly account for wind shear), while tube-based integral approximations provide a practical pathway to reduce the expense of full spatio-temporal 3D integrations.

Microphysical fidelity may be increased by augmenting the growth and habit-dynamics terms to include habit-specific ventilation corrections, aggregation and morphological evolution, e.g., hollowing or branching which directly modifies deposition density.

We suggest prioritizing the calibration of model parameters that carry the largest uncertainties by tuning against available contrail observations (for example, in-plume measurements or radiative-forcing diagnostics). Both theoretical analysis and numerical experiments indicate that the newly introduced ingredients, particularly bulk settling and explicit habit dynamics, can substantially influence integral plume properties. In particular (supported by the extensive set of simulations conducted throughout this study), the habit-resolving model exhibits notable absolute deviations in averaged ice water content (IWC) relative to a spherical-particle baseline. Nonetheless, both the sign and magnitude of this deviation may vary considerably depending on whether the ice crystals are in growth or sublimation phases. Moreover, it is hypothesized that the sign and magnitude are further modulated by the representation of more detailed habit morphologies and additional microphysical processes. Accordingly, more definitive conclusions must await further research in this area.

Overall, the proposed framework provides a computationally tractable, physically consistent basis for exploring such effects and for systematic model–data integration in future studies.

**Data Availability Statement.** Given that the research is theoretical in nature, there is no relevant data available for sharing.

**Conflict of Interest.** The authors declare no conflict of interest of any type within the present submission.

**Author contributions.** AJ conceptualized the research and developed the theoretical and computational framework. MS provided critical suggestions and improvements. AJ performed the numerical analyses. Both authors carefully reviewed the numerical results and carried out a preliminary sensitivity analysis with respect to the physical parameters. AJ and MS jointly contributed to the manuscript preparation, including initial drafting and multiple rounds of revision and improvement.



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

**Appendix**

**Appendix A: Composite Wind Field Construction**

- **Uniform flow:** the potential associated with a uniform flow is given by $\Phi^{(u)}(x,y) = U_\infty\, x + V_\infty\, y$, from which we obtain the velocity field as:

$$\mathbf{W}^{(u)}(x,y) = \nabla\Phi^{(u)} = \left(U_\infty, V_\infty\right). \tag{A1}$$

- **Regularized point vortices:** the stream function for each vortex of circulation $\Gamma_k$ at $(x_{0,k}, y_{0,k})$ with core radius $R_{0,k}^{(v)}$ is:

$$\Psi_k^{(v)}(x,y) = \frac{\Gamma_k}{4\pi}\ln\left[r_k^2 + \left(R_{0,k}^{(v)}\right)^2\right]. \tag{A2}$$

from which the velocity field is obtained by:

$$\mathbf{W}_k^{(v)}(x,y) = \nabla^\perp\Psi_k^{(v)} = \begin{pmatrix} -\partial_y\Psi_k^{(v)} \\ \partial_x\Psi_k^{(v)} \end{pmatrix} = \frac{\Gamma_k}{2\pi}\frac{1}{r_k^2 + \left(R_{0,k}^{(v)}\right)^2}\begin{pmatrix} -(y - y_{0,k}) \\ (x - x_{0,k}) \end{pmatrix}. \tag{A3}$$

where $r_k^2 = (x - x_{v,k})^2 + (y - y_{v,k})^2$.

- **Regularized dipoles:** each dipole of vector moment $\boldsymbol{\mu}_\ell = (\mu_{x,\ell}, \mu_{y,\ell})$ at $(x_{d,\ell}, y_{d,\ell})$ with regularization radius $R_{0,\ell}^{(d)}$ has potential:

$$\Phi_\ell(x,y) = \frac{\boldsymbol{\mu}_\ell\cdot\left(x - x_{d,\ell},\, y - y_{d,\ell}\right)}{2\pi\left[r_\ell^2 + \left(R_{0,\ell}^{(d)}\right)^2\right]}. \tag{A4}$$

from which we obtain the velocity field as:

$$\mathbf{W}_\ell^{(d)}(x,y) = \nabla\Phi_\ell =$$

$$\frac{1}{2\pi}\frac{1}{\left(r_\ell^2 + (R_{0,\ell}^{(d)})^2\right)^2}\begin{pmatrix} \mu_{x,\ell}\left[(x - x_{d,\ell})^2 - (y - y_{d,\ell})^2\right] + 2\,\mu_{y,\ell}\,(x - x_{d,\ell})(y - y_{d,\ell}) \\ \mu_{y,\ell}\left[(y - y_{d,\ell})^2 - (x - x_{d,\ell})^2\right] + 2\,\mu_{x,\ell}\,(x - x_{d,\ell})(y - y_{d,\ell}) \end{pmatrix}. \tag{A5}$$

where $r_\ell^2 = (x - x_{d,\ell})^2 + (y - y_{d,\ell})^2$.





- **_Regularized point sources/sinks:_** each source (positive strength) or sink (negative strength) of net strength $Q_j$ at $(x_{s,j}, y_{s,j})$ with core radius $R_{0,j}^{(s)}$ has potential:

$$\Phi_j^{(s)}(x,y) = \frac{Q_j}{4\pi} \ln\!\left[ r_j^2 + \left(R_{0,j}^{(s)}\right)^2 \right]. \tag{A6}$$

from which we obtain:

$$\mathbf{W}_j^{(s)}(x,y) = \nabla \Phi_j^{(s)} = \frac{Q_j}{2\pi} \frac{1}{r_j^2 + \left(R_{0,j}^{(s)}\right)^2} \begin{pmatrix} x - x_{s,j} \\ y - y_{s,j} \end{pmatrix}. \tag{A7}$$

where $r_j^2 = (x - x_{s,j})^2 + (y - y_{s,j})^2$.

In each case, the regularization radius $R_{0,k}^{(v)}$, $R_{0,\ell}^{(d)}$, and $R_{0,j}^{(s)}$ (for vortices, dipoles, and sources/sinks respectively) prevents singular behavior at the core. By construction, every component satisfies: $\nabla \cdot \mathbf{W}^{(u)} = 0$, $\nabla \cdot \mathbf{W}_k^{(v)} = 0$, $\nabla \cdot \mathbf{W}_\ell^{(d)} = 0$, $\nabla \cdot \mathbf{W}_j^{(s)} = 0$. Therefore, by linearity of divergence:

$$\nabla \cdot \mathbf{W}(x,y) = \nabla \cdot \left( \mathbf{W}^{(u)} + \sum_k \mathbf{W}_k^{(v)} + \sum_\ell \mathbf{W}_\ell^{(d)} + \sum_j \mathbf{W}_j^{(s)} \right) = 0. \tag{A8}$$

The proposed composite inviscid wind model is characterized by the following free parameters, which can be fine-tuned using the available wind data:

$$\textbf{Parameter Count} = \underbrace{U_\infty, V_\infty}_{2} + \underbrace{4M_v}_{\Gamma_k, (x_{v,k}, y_{v,k}), R_{0,k}^{(v)}} + \underbrace{5M_d}_{(\mu_{x,\ell}, \mu_{y,\ell}), (x_{d,\ell}, y_{d,\ell}), R_{0,\ell}^{(d)}} + \underbrace{4M_s}_{Q_j, (x_{s,j}, y_{s,j}), R_{0,j}^{(s)}}. \tag{A9}$$

## A1  Wind Turbulence Closure

For each horizontal component $i \in \{x, y\}$ we decompose:

$$w_i(x,y,t) = w_{i,m}(x,y,t) + w_i'(x,y,t), \tag{A10}$$

with turbulent fluctuations prescribed by:

$$w_i'(x,y,t) = \sigma_{w_i}(x,y,t) \Re\!\left\{ \mathcal{F}^{-1}\!\left[ \sqrt{E(k_x, k_y)}\, \xi_i(k_x, k_y, t) \right] \right\}, \tag{A11}$$

where we use the Fourier convention $\mathcal{F}^{-1}\{G\}(x,y) = \frac{1}{(2\pi)^2} \iint_{\mathbb{R}^2} G(k_x, k_y)\, e^{i(k_x x + k_y y)}\, dk_x dk_y$. The random fields $\xi_i(k_x, k_y, t)$ are complex-valued, zero-mean, unit-variance processes, and independent for $i \neq j$.

We set the local turbulence intensity:

$$I_{w_i}(x,y,t) = \frac{\sigma_{w_i}(x,y,t)}{|w_{i,m}(x,y,t)|}, \tag{A12}$$

The horizontal energy spectrum is taken as von Kármán:

$$E(k_x, k_y) = C \frac{k^4}{(k^2 + L^{-2})^{17/6}}, \qquad k = \sqrt{k_x^2 + k_y^2}, \tag{A13}$$

where $L$ is the integral scale and $C$ is the normalization constant.





## Appendix B: Derivation of Moment Equations from the Population Balance Equation

In this section, we derive the macroscopic advection–diffusion equations (ADEs) for the number and mass concentrations of ice particles from the Population Balance Equation (PBE).

### B1 Population Balance Equation (PBE)

The PBE for the distribution function $f(\mathbf{x}, m, t)$ is given by:

$$\frac{\partial f}{\partial t} + \nabla \cdot \left( \mathbf{v}_p\, f \right) = -\frac{\partial}{\partial m} \left( \dot{m}\, f \right) + \nabla \cdot \left( \tilde{\mathcal{D}}\, \nabla f \right) + S_f, \tag{B1}$$

where, $\mathbf{v}_p(\mathbf{x}, t)$ is the particle bulk velocity, defined as:

$$\mathbf{v}_p = \mathbf{v}_{slp} + \underbrace{(w_x, w_y, w_z)^\top}_{\substack{\text{background velocity} \\ \text{(wind components)}}} \approx (0, 0, v_s)^\top + \underbrace{(w_x, w_y, w_z)^\top}_{\text{background velocity}} = (w_x, w_y, w_z + v_s)^\top. \tag{B2}$$

In Eq. (B2), $\mathbf{v}_{slp}$ denotes the bulk slip velocity of the particle phase within a fluid undergoing turbulent mixing.

In addition, $\dot{m} = \rho_{\text{dep}}\, f_{\mathcal{V}}(m, \mathbf{x}, t)$ is the particle growth rate, $\tilde{\mathcal{D}}(\mathbf{x}, t, c_N, m)$ is the diffusion coefficient, and $S_f(\mathbf{x}, m, t)$ represents source/sink terms. Since the particle mass distribution is narrowly peaked, under the monodisperse assumption, at a representative mass $\bar{m}(\mathbf{x}, t)$, hence $f(\mathbf{x}, m, t) = c_N(\mathbf{x}, t)\, \delta\left( m - \bar{m}(\mathbf{x}, t) \right)$, where $c_N(\mathbf{x}, t)$ is the particle number concentration. Therefore, we can write: $f_{\mathcal{V}}(m, \mathbf{x}, t) \approx f_{\mathcal{V}}\left( \bar{m}(\mathbf{x}, t), \mathbf{x}, t \right)$, $\tilde{\mathcal{D}}(\mathbf{x}, m, t, c_N) \approx \tilde{\mathcal{D}}(\mathbf{x}, \bar{m}(\mathbf{x}, t), t, c_N)$, and $\mathbf{v}_p(\mathbf{x}, m, t) \approx \mathbf{v}_p(\mathbf{x}, \bar{m}(\mathbf{x}, t), t)$.

The number and mass concentrations are defined by:

$$c_N(\mathbf{x}, t) = \int_0^\infty f(\mathbf{x}, m, t)\, dm, \quad c_M(\mathbf{x}, t) = \int_0^\infty m\, f(\mathbf{x}, m, t)\, dm. \tag{B3}$$

Using the monodisperse assumption, we obtain:

$$c_N(\mathbf{x}, t) = c_N(\mathbf{x}, t), \quad c_M(\mathbf{x}, t) = \bar{m}(\mathbf{x}, t)\, c_N(\mathbf{x}, t). \tag{B4}$$

### B2 Derivation of the Zeroth Moment Equation

Integrating the PBE (B1) over $m$:

$$\int_0^\infty \left\{ \frac{\partial f}{\partial t} + \nabla \cdot \left( \mathbf{v}_p\, f \right) \right\} dm = \int_0^\infty \left\{ -\frac{\partial}{\partial m} \left( \dot{m}\, f \right) + \nabla \cdot \left( \tilde{\mathcal{D}}\, \nabla f \right) + S_f \right\} dm \ \Rightarrow$$

$$\frac{\partial}{\partial t} \int_0^\infty f\, dm + \left( \nabla \cdot \mathbf{v}_p \right) \int_0^\infty f\, dm + \mathbf{v}_p \cdot \nabla \int_0^\infty f\, dm = -[\dot{m}\, f]_0^\infty + \nabla \cdot \left( \tilde{\mathcal{D}}\, \nabla \int_0^\infty f\, dm \right) \ \Rightarrow$$

$$\frac{\partial c_N}{\partial t} + \nabla \cdot \left( \mathbf{v}_p\, c_N \right) = \nabla \cdot \left( \tilde{\mathcal{D}}\, \nabla c_N \right) + S_{c_N} \tag{B5}$$

where $S_{c_N}(\mathbf{x}, t) = \int_0^\infty S_f(\mathbf{x}, m, t)\, dm$. Notably, the boundary term $[\dot{m}\, f]_0^\infty$ cancels out since there is no particle distribution at $m = 0$ or as $m \to \infty$.





## B3 Derivation of the First Moment Equation

Multiplying the PBE by $m$ and integrating over $m$:

$$\int_0^\infty m\left\{\frac{\partial f}{\partial t} + \nabla\cdot(\mathbf{v}_p\, f)\right\}dm = \int_0^\infty m\left\{-\frac{\partial}{\partial m}(\dot{m}\,f) + \nabla\cdot(\tilde{\mathcal{D}}\nabla f) + S_f\right\}dm \;\Rightarrow$$

$$\frac{\partial}{\partial t}\int_0^\infty m\,f\,dm + (\nabla\cdot\mathbf{v}_p)\int_0^\infty m\,f\,dm + \mathbf{v}_p\cdot\nabla\int_0^\infty m\,f\,dm =$$

$$-[m\,\dot{m}\,f]_0^\infty + \int_0^\infty \dot{m}\,f\,dm + \nabla\cdot\left(\tilde{\mathcal{D}}\nabla\int_0^\infty m\,f\,dm\right) \;\Rightarrow$$

$$\frac{\partial c_M}{\partial t} + \nabla\cdot(\mathbf{v}_p\, c_M) = \nabla\cdot\left(\tilde{\mathcal{D}}\nabla c_M\right) + \rho_{dep}\, f_{\mathcal{V}}\big(\bar{m}(\mathbf{x},t),\mathbf{x},t\big)\, c_N(\mathbf{x},t) + S_{c_M}. \tag{B6}$$

where $S_{c_M}(\mathbf{x},t) = \int_0^\infty m\, S_f(\mathbf{x},m,t)\,dm$.

## B4 Derivation of the Per-Particle Mass Evolution Equation

In previous sections, the representative ice-particle mass was denoted by $\bar{m}$; here, by slight abuse of notation we write $m$ for the same quantity. We omit all source terms except the growth term $\rho_{\text{dep}}\, f_{\mathcal{V}}$ in the $c_M$ equation. Under monodispersity, we have $m = \frac{c_M}{c_N}$. Combining the two moments, we write:

$$\left[\frac{\partial c_M}{\partial t} + \nabla\cdot(\mathbf{v}_p\, c_M)\right] - m\left[\frac{\partial c_N}{\partial t} + \nabla\cdot(\mathbf{v}_p\, c_N)\right]$$

$$= \nabla\cdot\left(\tilde{\mathcal{D}}\nabla c_M\right) - m\,\nabla\cdot\left(\tilde{\mathcal{D}}\nabla c_N\right) + \rho_{dep}\, f_{\mathcal{V}}\, c_N. \tag{B7}$$

Since:

$$\frac{\partial(c_N m)}{\partial t} = m\frac{\partial c_N}{\partial t} + c_N\frac{\partial m}{\partial t}, \tag{B8}$$

$$\nabla\cdot(\mathbf{v}_p\, m\, c_N) = c_N\,\mathbf{v}_p\cdot\nabla m + m\,\mathbf{v}_p\cdot\nabla c_N + m\,c_N\,\nabla\cdot\mathbf{v}_p, \tag{B9}$$

the $m$-weighted terms cancel upon subtraction and the left-hand side reduces to:

$$c_N\left(\frac{\partial m}{\partial t} + \mathbf{v}_p\cdot\nabla m\right) = c_N\frac{\mathrm{D}m}{\mathrm{D}t}. \tag{B10}$$

Thus:

$$c_N\frac{\mathrm{D}m}{\mathrm{D}t} = \nabla\cdot\left(\tilde{\mathcal{D}}\nabla(c_N m)\right) - m\,\nabla\cdot\left(\tilde{\mathcal{D}}\nabla c_N\right) + \rho_{dep}\, f_v\, c_N. \tag{B11}$$

Moreover, we have:

$$\nabla(c_N m) = c_N\nabla m + m\nabla c_N, \tag{B12}$$





$$\nabla^2(c_N m) = c_N \nabla^2 m + 2\,\nabla m \cdot \nabla c_N + m\nabla^2 c_N, \tag{B13}$$

Therefore, expanding the diffusion difference gives:

$$\nabla\cdot\big(\tilde{\mathcal{D}}\nabla(c_N m)\big) - m\nabla\cdot\big(\tilde{\mathcal{D}}\nabla c_N\big)$$
$$= \tilde{\mathcal{D}}\big(c_N\nabla^2 m + 2\,\nabla m\cdot\nabla c_N + m\nabla^2 c_N\big) + (\nabla\tilde{\mathcal{D}})\cdot(c_N\nabla m + m\nabla c_N)$$
$$\quad - m\Big(\tilde{\mathcal{D}}\nabla^2 c_N + (\nabla\tilde{\mathcal{D}})\cdot\nabla c_N\Big)$$
$$= c_N\big[\tilde{\mathcal{D}}\nabla^2 m + (\nabla\tilde{\mathcal{D}})\cdot\nabla m\big] + 2\tilde{\mathcal{D}}\nabla m\cdot\nabla c_N. \tag{B14}$$

Hence, using $\tilde{\mathcal{D}}\nabla^2 m + (\nabla\tilde{\mathcal{D}})\cdot\nabla m = \nabla\cdot(\tilde{\mathcal{D}}\nabla m)$ the per-particle evolution equation in material form is written as:

$$\frac{\partial m}{\partial t} + \mathbf{v}_p\cdot\nabla m = \nabla\cdot\big(\tilde{\mathcal{D}}\nabla m\big) + \frac{2\tilde{\mathcal{D}}}{c_N}\nabla m\cdot\nabla c_N + \rho_{\mathrm{dep}}\,f_{\mathcal{V}}. \tag{B15}$$

**Appendix C: Derivation of the Particle-Phase Momentum Equation**

In multiphase flow modeling, the Euler–Euler framework treats both the fluid and particulate phases as interpenetrating continua. To better conceptualize this, consider the particle phase settling within a fluid subject to turbulent mixing. Since the mean fluid motion (i.e., wind components) is already incorporated into the definition of $\mathbf{v}_p$, the fluid velocity appearing in the slip velocity $\mathbf{v}_{\mathrm{slp}}$ accounts only for turbulent fluctuations. Under this setup, the full momentum

equation for the particle phase takes the following form (Brennen, 2005):

$$\frac{\partial(\epsilon_p\rho_p\mathbf{v}_{slp})}{\partial t} + \nabla\cdot\big(\epsilon_p\rho_p\,\mathbf{v}_{slp}\mathbf{v}_{slp}\big) = -\epsilon_p\nabla p + \nabla\cdot\tau_p + \epsilon_p\rho_p\,\mathbf{g} + F_d, \tag{C1}$$

where $\epsilon_p$ is the particle volume fraction, $\rho_p$ is the particle density, $\mathbf{v}_{slp}$ is the particle bulk slip velocity, $p$ is the pressure due to the fluid phase, $\tau_p$ is the particle-phase stress tensor (namely deviatoric steress emerging on the exterior of the control volume), $\mathbf{g}$ is the gravitational acceleration vector, and $F_d$ is the net force per unit volume acting on the

particles in the control volume. In many closures the net force is modeled as proportional to $\epsilon_p$ (e.g., arising from the per-particle contribution multiplied by the local particle concentration, i.e., $F_d = \epsilon_p f_d$ where $f_d$ is the average net force experienced by individual particles which is inherently different from the drag force experienced by a single settling particle in an unbounded domain). Local mechanical equilibrium $p_p = p_f$ is assumed, so that the pressure gradient acting on the particle phase is $-\epsilon_p\nabla p$. The stress tensor can be expressed as: $\tau_p = \epsilon_p\mu_{t,p}(\nabla\mathbf{v}_{slp} + \nabla\mathbf{v}_{slp}^\top)$.



Expanding the LHS of Eq. (C1), we write:

$$\frac{\partial(\epsilon_p \rho_p \mathbf{v}_{slp})}{\partial t} + \nabla \cdot \left(\epsilon_p \rho_p \mathbf{v}_{slp}\mathbf{v}_{slp}\right)$$

$$= \rho_p \mathbf{v}_{slp}\frac{\partial \epsilon_p}{\partial t} + \epsilon_p \rho_p \frac{\partial \mathbf{v}_{slp}}{\partial t} + \rho_p \mathbf{v}_{slp}\nabla \cdot (\epsilon_p \mathbf{v}_{slp}) + \epsilon_p \rho_p (\mathbf{v}_{slp} \cdot \nabla)\mathbf{v}_{slp}$$

$$= \rho_p \mathbf{v}_{slp}\underbrace{\left[\frac{\partial \epsilon_p}{\partial t} + \nabla \cdot (\epsilon_p \mathbf{v}_{slp})\right]}_{\substack{\text{continuity equation of}\\\text{the particle phase}}} + \epsilon_p \rho_p \left[\frac{\partial \mathbf{v}_{slp}}{\partial t} + (\mathbf{v}_{slp} \cdot \nabla)\mathbf{v}_{slp}\right]. \tag{C2}$$

For the stress term, we write:

$$\frac{1}{\epsilon_p \rho_p}\nabla \cdot \tau_p = \frac{\nu_t}{\epsilon_p}\nabla \cdot [\epsilon_p(\nabla \mathbf{v}_{slp} + \nabla \mathbf{v}_{slp}^\top)] = \nu_t\left[\nabla^2 \mathbf{v}_{slp} + \frac{1}{\epsilon_p}(\nabla \epsilon_p \cdot \nabla)\mathbf{v}_{slp} + \frac{1}{\epsilon_p}(\nabla \epsilon_p \cdot \nabla)^\top \mathbf{v}_{slp}\right] \tag{C3}$$

where $\nu_t := \frac{\mu_{t,p}}{\rho_p}$ is the average turbulent eddy-viscosity.

Substituting the stress term into Eq. (C1) and invoking the first moment equation, Eq. (B6), to represent particle-phase continuity, where $c_M = \rho_{dep}\epsilon_p = \rho_p \epsilon_p$, $\nabla \cdot \mathbf{v}_{slp} \approx 0$, and no additional particle injection is assumed (i.e., $S_{c_M} = 0$), while also noting that the advection term has been replaced with $\mathbf{v}_{slp}$, we obtain:

$$\frac{\partial \mathbf{v}_{slp}}{\partial t} + (\mathbf{v}_{slp} \cdot \nabla)\mathbf{v}_{slp} = \nu_t\left[\nabla^2 \mathbf{v}_{slp} + \frac{1}{\epsilon_p}(\nabla \epsilon_p \cdot \nabla)\mathbf{v}_{slp} + \frac{1}{\epsilon_p}(\nabla \epsilon_p \cdot \nabla)^\top \mathbf{v}_{slp}\right] - \frac{\nabla p}{\rho_p} + \mathbf{g} + \frac{f_d}{\rho_p} -$$

$$\frac{\mathbf{v}_{slp}}{\epsilon_p}\left[\nabla \cdot \left(\tilde{\mathcal{D}}\nabla \epsilon_p\right) + f_v\big(\bar{m}(\mathbf{x},t),\mathbf{x},t\big)c_N(\mathbf{x},t)\right]. \tag{C4}$$

In general, the above equation is coupled with the fluid phase through the pressure and force terms, and with the
first and second moments of the particle distribution. Notably, as $f_d$ approaches the drag force on a single settling particle in an unbounded domain, the pressure gradient tends toward the hydrostatic condition, $\nabla p = \rho_f \mathbf{g}$.

The above equation can be written as:

$$\frac{\partial \mathbf{v}_{slp}}{\partial t} + (\mathbf{v}_{slp} \cdot \nabla)\mathbf{v}_{slp} = \nu_t \nabla^2 \mathbf{v}_{slp} + \boldsymbol{C}_f + \boldsymbol{C}_{c_M, c_N, f_v}. \tag{C5}$$

where $\boldsymbol{C}_f$ represents the coupling between $\mathbf{v}_{slp}$ and the fluid phase, and is expressed as $\boldsymbol{C}_f = -\frac{\nabla p}{\rho_p} + \mathbf{g} + \frac{f_d}{\rho_p}$; and
$\boldsymbol{C}_{c_M, c_N, f_v}$ captures the coupling of $\mathbf{v}_{slp}$ to the mass and number concentration fields, as well as to the growth term, and is given by $\boldsymbol{C}_{c_M, c_N, f_v} = -\frac{\mathbf{v}_{slp}}{\epsilon_p}\left[\nabla \cdot \left(\tilde{\mathcal{D}}\nabla \epsilon_p\right) + f_v\big(\bar{m}(\mathbf{x},t),\mathbf{x},t\big)c_N(\mathbf{x},t)\right] + \nu_t\left[\frac{1}{\epsilon_p}(\nabla \epsilon_p \cdot \nabla)\mathbf{v}_{slp} + \frac{1}{\epsilon_p}(\nabla \epsilon_p \cdot \nabla)^\top \mathbf{v}_{slp}\right]$.

Accurate solution of the above equation presents formidable challenges in (i) specifying boundary and initial conditions, (ii) coupling the evolving fluid phase with particle number and mass concentrations, and (iii) incorporating microphysical growth rates. In addition, in practice, the eddy viscosity is not only space- and time-dependent, but
also anisotropic-varying across different spatial directions.

## A Compressed Low-Order Model

The above equation can be expressed solely in terms of the particle bulk settling velocity $v_s$ in the $z$-direction by collapsing the turbulent-mixing in the vertical dimension. In other words, the 'quasi-hovering' (*loitering*) behavior of





horizontally distributed particles can be interpreted as an effective turbulent mixing in the vertical direction. For this
reason, this type of 1D modeling is referred to as a *compressed low-order model*.

This framework motivates defining an effective vertical eddy viscosity, $\nu_{\mathbf{t,ef}} \approx \langle \nu_t(x,y,z) \rangle$, which encapsulates the total turbulent mixing effect.

Here, we assume that the particle bulk velocity is initially zero at a reference plane, $z_{\mathbf{ref}}$, and remains negligible for an extended period due to strong early-time turbulent mixing effect when the concentration field is at its maximum.
Far below $z_{\mathbf{ref}}$, the local concentration decreases, and particle growth amplifies gravitational settling. This leads to an increase in both the Galileo and Stokes numbers. Consequently, the bulk net force $f_d$ asymptotically converges to the drag force experienced by an individual particle in an unbounded domain, and the pressure gradient approaches hydrostatic balance, $\nabla p = \rho_f \mathbf{g}$.

Under this framework, the fluid coupling term $\boldsymbol{C}_f$ is encoded within the boundary conditions, and, for simplicity,
the term representing coupling to the concentration field, $\boldsymbol{C}_{c_M,c_N,f_v}$, is not explicitly accounted for. However, since the present bulk settling model is coupled to the microphysical ADEs, the particle growth term is implicitly included, as the growth makes $v_{\mathbf{ter}} := v_{\mathbf{ter}}(z,t)$.

Within this framework, the fluid coupling term $\boldsymbol{C}_f$ is encoded within the boundary conditions. For simplicity, the coupling term associated with the concentration field, $\boldsymbol{C}_{c_M,c_N,f_v}$, is neglected. Nonetheless, because the bulk settling
model is coupled to the microphysical ADEs, the effects of particle growth are implicitly represented through the time dependence of the terminal velocity, $v_{\mathbf{ter}} := v_{\mathbf{ter}}(z,t)$. Therefore, with appropriately chosen boundary conditions, Eq. (C5) reduces to the following 1D model:

$$\frac{\partial v_s}{\partial t} + v_s \frac{\partial v_s}{\partial z} = \nu_{\mathbf{t,ef}} \frac{\partial^2 v_s}{\partial z^2}, \quad v_s(z_{\mathbf{ref}},t=0)=0, \quad v_s(z \ll z_{\mathbf{ref}}, t=0)=v_{\mathbf{ter}}, \quad \lim_{t\to\infty} v_s = v_{\mathbf{ter}}. \tag{C6}$$

where $v_{\mathbf{ter}}$ denotes the conventional terminal velocity of individual particles in an unbounded domain. Notably, by
prescribing $v_{\mathbf{ter}}$ as the outer boundary condition, the model inherently captures the interplay between (i) turbulent mixing in the highly concentrated upper region (*loitering*), (ii) the transition of the bulk net force $f_d$ into the drag force on individual particles, and (iii) the acceleration of this transition by micro-physical growth processes.

In addition, by expressing the asymptotic settling velocity as $\lim_{t\to\infty} v_s = \alpha v_{\mathbf{ter}}$, where $\alpha \propto \mathrm{St}$ or $\mathrm{Ga}$, the model more accurately captures the *sweeping* mechanism. By allowing $\alpha$ to exceed unity in inertia-dominated regimes, this
formulation can represent the increased settling velocity associated with the *sweeping* mechanism.

## Appendix D: Closed-Form Solution for the Bulk Settling Velocity $v_s(z,t)$

Employing the Cole–Hopf transformation,

$$\hat{v}_s(z,t) = -2\nu_{\mathbf{t,ef}} \frac{\partial}{\partial z} \ln \varphi(z,t) = -2\nu_{\mathbf{t,ef}} \frac{\frac{\partial \varphi}{\partial z}}{\varphi}. \tag{D1}$$



where $\varphi$ satisfies the linear heat equation:

$$\frac{\partial \varphi}{\partial t} = \nu_{\mathbf{t,ef}} \frac{\partial^2 \varphi}{\partial z^2}. \tag{D2}$$

with the initial condition $\varphi(z,0) =: \varphi_0(z)$ as:

$$\varphi_0(z) = \exp\left(-\frac{1}{2\nu_{\mathbf{t,ef}}} \int_{z_{\mathbf{relax}}}^{z} \hat{v}_s(\xi,0)\,d\xi\right) = \begin{cases} \exp\left(-\frac{v_{\mathbf{ter}}}{2\nu_{\mathbf{t,ef}}}(z - z_{\mathbf{relax}})\right), & z \leq z_{\mathbf{relax}}, \\ 1, & z > z_{\mathbf{relax}}. \end{cases} \tag{D3}$$

The solution of the heat equation with initial data $\varphi_0$ is given by convolution with the heat kernel:

$$G(\xi,t) = \frac{1}{\sqrt{4\pi\nu_{\mathbf{t,ef}}t}} \exp\left(-\frac{\xi^2}{4\nu_{\mathbf{t,ef}}t}\right), \qquad t > 0, \tag{D4}$$

so for $t > 0$

$$\varphi(z,t) = \int_{-\infty}^{\infty} G(z-y,t)\,\varphi_0(y)\,dy. \tag{D5}$$

Splitting the integral at $y = z_{\mathbf{relax}}$, and setting $\Delta_0 := z - z_{\mathbf{relax}}$, we write:

$$\varphi(z,t) = \int_{-\infty}^{z_{\mathbf{relax}}} G(z-y,t)\,e^{-\frac{v_{\mathbf{ter}}}{2\nu_{\mathbf{t,ef}}}(y-z_{\mathbf{relax}})}\,dy \;+\; \int_{z_{\mathbf{relax}}}^{\infty} G(z-y,t)\,dy =: I_1 + I_2. \tag{D6}$$

$I_2$ can be computed as:

$$I_2 = \frac{1}{\sqrt{4\pi\nu_{\mathbf{t,ef}}t}} \int_{z_{\mathbf{relax}}}^{\infty} \exp\left(-\frac{(z-y)^2}{4\nu_{\mathbf{t,ef}}t}\right) dy = \frac{1}{2}\left(1 + \mathrm{erf}\left(\frac{\Delta_0}{2\sqrt{\nu_{\mathbf{t,ef}}t}}\right)\right). \tag{D7}$$

For $I_1$, we define $s := y - z_{\mathrm{relax}}$ so $s \in (-\infty, 0]$. Thus:

$$I_1 = \frac{1}{\sqrt{4\pi\nu_{\mathbf{t,ef}}t}} \int_{-\infty}^{0} \exp\left(-\frac{(\Delta_0 - s)^2}{4\nu_{\mathbf{t,ef}}t} - \frac{v_{\mathbf{ter}}}{2\nu_{\mathbf{t,ef}}}s\right) ds. \tag{D8}$$

which can be written as:

$$I_1 = \exp\left(\frac{v_{\mathbf{ter}}^2 t}{4\nu_{\mathbf{t,ef}}} - \frac{v_{\mathbf{ter}}\Delta_0}{2\nu_{\mathbf{t,ef}}}\right) \cdot \frac{1}{\sqrt{4\pi\nu_{\mathbf{t,ef}}t}} \int_{-\infty}^{0} \exp\left(-\frac{(s - (\Delta_0 - v_{\mathbf{ter}}t))^2}{4\nu_{\mathbf{t,ef}}t}\right) ds =$$

$$\exp\left(\frac{v_{\mathbf{ter}}^2 t}{4\nu_{\mathbf{t,ef}}} - \frac{v_{\mathbf{ter}}\Delta_0}{2\nu_{\mathbf{t,ef}}}\right) \cdot \frac{1}{2}\left(1 + \mathrm{erf}\left(\frac{v_{\mathbf{ter}}t - \Delta_0}{2\sqrt{\nu_{\mathbf{t,ef}}t}}\right)\right). \tag{D9}$$

Therefore, $\varphi(z,t)$ is obtained as $\varphi(z,t) = I_1 + I_2$, and recovering $\hat{v}_s$ using (D1), gives:

$$\hat{v}_s(z,t) = v_{\mathbf{ter}} \frac{\big(1 + \mathrm{erf}(p_0)\big)\,p_2}{\big(1 + \mathrm{erf}(p_0)\big)\,p_2 + \big(1 + \mathrm{erf}(p_1)\big)} \tag{D10}$$

where:

$$p_0 := \frac{v_{\mathbf{ter}}t - \Delta_0}{2\sqrt{\nu_{\mathbf{t,ef}}t}}, \qquad p_1 := \frac{\Delta_0}{2\sqrt{\nu_{\mathbf{t,ef}}t}}, \qquad p_2 := \exp\left(\frac{v_{\mathbf{ter}}^2 t}{4\nu_{\mathbf{t,ef}}} - \frac{v_{\mathbf{ter}}\Delta_0}{2\nu_{\mathbf{t,ef}}}\right). \tag{D11}$$

The above equation can be also written in a more compact form by using the identity $(1 + \mathrm{erf}(\xi)) = \mathrm{erfc}(-\xi)$.



## Appendix E: Inherent Growth Factor $\Gamma^*$

It is worth emphasizing that our current habit-dynamics framework does not account for riming processes, nor does it explicitly represent bullet-rosette or hollowing/branching morphologies. Incorporating these features would introduce additional poorly constrained parameters, and the experimental literature for temperatures below $-40\,°C$ remains too sparse to support their robust parameterization.

The available mathematical frameworks for habit dynamics are those by Chen and Lamb (Chen and Lamb, 1994) (based on the aspect-ratio hypothesis, defining $\frac{dc}{da} = \frac{\alpha_c}{\alpha_a}\frac{c}{a} := \Gamma\phi$) and Nelson and Baker (Nelson and Baker, 1996) (based on the facet hypothesis, defining $\frac{dc}{da} = \frac{\alpha_c}{\alpha_a} := \Gamma$), where $\alpha_a$ and $\alpha_c$ are the deposition coefficients of the basal and prism axes, respectively, $a$ is half the basal-plane maximum width (equatorial radius), $c$ is half the prism-plane height (transverse radius), and $\Gamma$ is known as the Inherent Growth Factor (IGF). It is straightforward to show that the above dynamics are equivalent to:

$$\frac{d\phi}{d\mathcal{V}} = \frac{\Gamma^* - 1}{\Gamma^* + 2}\frac{d\phi}{d\mathcal{V}}, \tag{E1}$$

where $\Gamma^* = \Gamma$ in the Chen and Lamb model, and $\Gamma^* = \frac{\Gamma}{\phi}$ in the Nelson and Baker model.

For temperatures above $243.15\,K$, $\Gamma = \Gamma(T)$ in the Chen and Lamb model, and can be directly obtained by fitting a function to the IGF from the available experimental data in (Chen and Lamb, 1994; Zhang and Harrington, 2014). However, this range barely overlaps with the range through which contrails form and persist, and, as mentioned, the experimental values for deposition coefficients are too sparse for $T < 233.15\,K$.

In this research we set $\Gamma = \Gamma(T)$ for $T > 243.15\,K$ and directly fit a function $\Gamma(T)$ to the available data in (Chen and Lamb, 1994; Zhang and Harrington, 2014). However, for $T < 243.15\,K$ we use Nelson and Baker model which directly accounts for dislocation growth and step nucleation theories, characterized by supersaturation immediately above the surface (i.e., surface supersaturation $s_{\mathrm{surf}}$), a temperature-dependent characteristic supersaturation describing the supersaturation-dependence of surface-kinetic mediated growth (i.e., $s_{\mathrm{char}}$), and the parameter $M$ that describes the surface growth mode. An approximation for $\alpha$ the captures both dislocation growth and step nucleation theories was suggested by Nelson and Baker (Nelson and Baker, 1996):

$$\alpha(T, s_i) = \alpha_s \left(\frac{s_{\mathrm{surf}}}{s_{\mathrm{char}}}\right)^M \tanh\left(\frac{s_{\mathrm{char}}}{s_{\mathrm{surf}}}\right)^M \tag{E2}$$

where $\alpha_s$ is the sticking probability/adsorption efficiency and is thought to be near unity (Harrington et al., 2021; Laurencin et al., 2022). In addition, $M = 1$ is consistent with the dislocation growth whereas $M \geq 10$ is amenable to step nucleation.

As discussed earlier, mainly because there is not a general agreement in habit modeling below $T < 233K$ (interested readers are referred to (Pruppacher and Klett, 1997; Harrington et al., 2021; Laurencin et al., 2022; Harrington et al., 2019; Harrington and Pokrifka, 2024)), in this research, we also partially rely on the ice-crystal habit diagrams in the $-40\,°C$ to $-70\,°C$ range (e.g.(Bailey and Hallett, 2010, 2009)); indicating that at very low vapor supersaturations, below approximately 5%, crystals often grow into thick, irregular platelets, effectively decoupling habit form from temperature.





In this low-supersaturation regime, the shape factor $\Gamma$ also becomes sensitive to variations in background pressure. In addition, deposition coefficients for $T < -40\,^\circ\mathrm{C}$ are largely unmeasured and carry substantial uncertainty. Therefore, here, we introduce an ad hoc modification (mimicking the 2D-nucleation–limited growth (Zhang and Harrington, 2014; Kuroda and Lacmann, 1982)) to enforce a kinetic-barrier–driven roll-off of $\Gamma$ as $s_i \to 0$. The characteristic surface supersaturation $s_{\mathrm{char}} = s_{\mathrm{char}}(T)$ is specified using the Nelson–Baker/Harrington parameterization (Harrington et al., 2021). Because laboratory constraints on $s_{\mathrm{char}}$ are sparse below approximately $-30\,^\circ\mathrm{C}$ (243.15 K), we fit smooth functions to the available experimental range and extrapolate these fits for $T < 243.15$ K. Following (Harrington et al., 2021), we compute $s_{\mathrm{surf}}$ as:

$$s_{\mathrm{surf}} \approx s_{\mathrm{diff}}^{1-\beta}\, s_{\mathrm{char}}^{\beta}. \tag{E3}$$

where:

$$s_{\mathrm{diff,a}} = \frac{s_i}{1 + L_a}, \quad L_a = \frac{a\,c\,\bar{v}_v}{4 D_v C_\Delta(c,a)}, \quad s_{\mathrm{diff,c}} = \frac{s_i}{1 + L_c}, \quad L_c = \frac{a^2\,\bar{v}_v}{4 D_v C_\Delta(c,a)}. \tag{E4}$$

where $D_v$ is the vapor diffusivity in air $\left(D_v = D_{v0}\left(\frac{T}{T_0}\right)^{1.94}\frac{p_0}{p}, D_{v0} = 2.11 \times 10^{-5}\ \mathrm{m^2 s^{-1}}, T_0 = 273.15\ \mathrm{K}, P_0 = 101325\right.$ Pa), $\bar{v}_v$ is the mean speed of a vapor molecule $\left(\bar{v} = \sqrt{\frac{8 k_B T}{\pi m}} = \sqrt{\frac{8 R T}{\pi M}},\right)$, where $k_B$ is Boltzmann's constant, $m$ the mass of one molecule, $R$ the universal gas constant and $M$ the molar mass. Moreover, $L_a$, and $L_c$ are unitless quantities that depend on the crystal geometry, and $C_\Delta$ is the capacitance (Westbrook et al., 2008) evaluated one mean free path from the surface, i.e., $C_\Delta := C(a+\Delta, c+\Delta)$ where $\Delta$ is approximately the mean free path. In addition, $\beta$ is a function of $M$ which can be directly obtained from (Harrington et al., 2021). Moreover, $s_{\mathrm{diff}}$, is the surface supersaturation over the respective axis when the deposition coefficient is unity.

Notably, modifications to the above formulas have been proposed, mainly for conditions of low ice supersaturation (see (Harrington et al., 2021)). In the present work, however, this regime is already treated through a blending function that enables the explicit formation of plate-like crystals, consistent with both the ice-crystal habit diagrams and observations of plate-like crystals in contrail cirrus. For this reason, we do not incorporate the residual modifications in our formulation. On choosing a critical supersaturation parameter $s_{\mathrm{cr.}}$, and a corresponding $\Gamma_{\mathrm{cr.}}$, the final form of the IGF for $T < 243.15 K$ is considered as:

$$\Gamma(T, s_i) = \Gamma_{\mathrm{cr.}} + \left(\frac{\alpha_c}{\alpha_a} - \Gamma_{\mathrm{cr.}}\right) e^{-\left(\frac{s_{\mathrm{cr.}}}{s_i}\right)^n}. \tag{E5}$$

According to the ice-crystal habit diagram we choose $s_{\mathrm{cr.}} \approx 5\%$, and propose $\Gamma_{\mathrm{cr.}} \in [0.2, 0.5]$, and $n \in [2, 4]$.



## Appendix F: Vapor Deposition Volumetric Growth Rate $f_\mathcal{V}$

The volumetric growth rate $f_\mathcal{V}$ is computed from a standard diffusion-limited growth model with latent-heat (thermal-resistance) coupling:

$$f_\mathcal{V} = \frac{4\pi C\, s_i}{\rho_{\text{dep}}\left(\dfrac{R_v T}{e_i D_v'} + \dfrac{L_s}{k_{\text{air}}' T}\left(\dfrac{L_s}{R_v T} - 1\right)\right)}. \tag{F1}$$

where $e_i = e_i(T)$ denotes the saturation vapor pressure over ice, and $L_s$ is the latent heat of sublimation. Empirical formulas from (Murphy and Koop, 2005) are used to evaluate $e_i$ and $L_s$. In addition, $R_v$ represents the specific gas

constant for water vapor, with a value of approximately $461.5\ \text{J kg}^{-1}\text{K}^{-1}$. In the absence of hollowing and branching processes, the deposition density $\rho_{\text{dep}}$ is approximated by the ice density, i.e., $\rho_{\text{dep}} \approx \rho_{\text{ice}} \approx 917\ \text{kg m}^{-3}$. Moreover, $D_v'$ denotes the kinetically limited vapor diffusivity, and $k_{\text{air}}'$ the kinetically limited thermal conductivity, expressed as (Zhang and Harrington, 2014; Pruppacher and Klett, 1997):

$$D_v' = \frac{2}{3}\frac{D_v}{A + B_d} + \frac{1}{3}\frac{D_v}{A + C_d}, \qquad k_{\text{air}}' = \frac{2}{3}\frac{k_{\text{air}}}{A + B_k} + \frac{1}{3}\frac{k_{\text{air}}}{A + C_k}. \tag{F2}$$

where:

$$A = \frac{C}{C_\Delta}, \quad B_d = \frac{4 D_v C}{\alpha_a v_v a^2 \varphi}, \quad C_d = \frac{4 D_v C}{\alpha_a \Gamma v_v a^2}, \quad B_k = \frac{4 k_{\text{air}} C}{0.7\,\rho_{\text{air}} c_p v_v a^2 \varphi}, \quad C_k = \frac{4 k_{\text{air}} C}{0.7\,\rho_{\text{air}} c_p v_v a^2}. \tag{F3}$$

In above, $k_{\text{air}}$ is the thermal conductivity of air ($\approx 0.024\ \text{W m}^{-1}\ \text{K}^{-1}$), $\rho_{\text{air}}$ is air density and $c_p$ is the specific heat capacity of air ($\approx 1005\ \text{J kg}^{-1}\text{K}^{-1}$). In addition, standard atmospheric relations are used to compute the air density, $\rho_{\text{air}}$, and pressure, $P$, from the temperature profile $T(z)$, where $T(z)$ follows the International Standard Atmosphere

lapse rate $-6.5 \times 10^{-3}\ \text{K m}^{-1}$.

## Appendix G: Diffusion Coefficient

The effective diffusion coefficient in contrail-plume models is state- and history-dependent and should be treated as nonlinear rather than constant. It is governed by finite particle loading (hindrance), particle morphology and aggregate porosity, collision-induced adhesion and aggregation, and flow-dependent dispersion (turbulence and shear).

For plume-scale simulations we therefore adopt a compact nonlinear parametrization that modifies a background diffusivity to represent loading and aggregation effects, capturing the dominant diffusion-blocking physics:

$$\mathcal{D}(c_N) \propto \frac{1}{1 + \beta c_N}. \tag{G1}$$

where $\beta$ is the diffusion-blocking constant

In this work we consider a diagonal stochastic diffusion tensor where,

$$\tilde{\mathcal{D}}(\mathbf{x}, t, c_N) = \begin{pmatrix} \tilde{\mathcal{D}}_{xx}(\mathbf{x}, t, c_N) & 0 & 0 \\ 0 & \tilde{\mathcal{D}}_{yy}(\mathbf{x}, t, c_N) & 0 \\ 0 & 0 & \tilde{\mathcal{D}}_{zz}(\mathbf{x}, t, c_N) \end{pmatrix}. \tag{G2}$$





where the deterministic parts are:

$$\mathcal{D}_{xx} := \frac{d_{0,x}}{1 + \beta_x c_N}, \quad \mathcal{D}_{yy} := \frac{d_{0,y}}{1 + \beta_y c_N}, \quad \mathcal{D}_{zz} := \frac{d_{0,z}}{1 + \beta_z c_N}. \tag{G3}$$

Ornstein–Uhlenbeck SDE can be employed to incorporate stochastic fluctuations into the effective diffusivities.