# Peer review of "A multi-physics Eulerian framework for long-term contrail evolution"

_EGUsphere, 2025_

## Referee Comment (RC2)

**Review of contrail study by Jafarimoghaddam & Soler subm. to ACP**

**Synthesis**

The manuscript describes a new modelling framework for simulations of the long-term contrail evolution. The manuscript is hard to read because many details are presented in an unstructured way (at least in my perception). It is verbose and repetitive in describing some model aspects, while other aspects that have been shown to be crucial by other modelling groups in the past are not well explained.

Even though the authors claim that advanced numerical schemes are employed, several a-priori modelling assumptions impose strong conceptual limitations. Moreover, some further odd choices do not boost my confidence in how the model is designed and reveal drawbacks, particularly in the microphysical approach.

The introduction claims that the new model uses more sophisticated representations of ice crystal habit and sedimentation.

Concerning the former aspect: I have strong doubts about the way you include ice crystal habits in your model and whether that approach is reasonable.

Concerning the latter aspect: The use of monodisperse size distributions does not allow to realistically model sedimentation with your model. This could obscure your findings about the relevance of an improved physical description of sedimentation.

Furthermore, the study should show how the modelling assumptions taken relate to more commonly used approaches. The study would also benefit from comparisons with other contrail-cirrus models. Not much connection is made to existing contrail-cirrus modelling studies and their findings.

It remains fuzzy whether the contrail evolutions you observe are similar to those shown in previous studies. This makes it also hard to figure out whether or not your model "does a good job".

**Major comments**

**1. Conciseness of text**

First of all, I would like to note that the manuscript is difficult to read because it addresses many aspects that I have not encountered in the contrail literature. I greatly appreciate that concepts, phenomena etc. from other research fields are being incorporated into the field of contrail research. Although these ideas may be common in other disciplines, it is unclear whether all of them are of primary importance for contrail evolution. You must clearly demonstrate that your novel model components are essential for contrail evolution.

In my view, the text is verbose and contains numerous technical details that should be relocated to the appendix or to a supplement. It is difficult to discern which aspects of the model and its assumptions are truly significant. Other essential information—such as the assumptions regarding a monodisperse size distribution—is buried deep within the text. The manuscript requires a much clearer structure and hierarchy regarding where which type of information is provided. Because many technical details appear in the main body, it is challenging to find the most essential information.

**2. Monodisperse assumption in microphysical modeling**

A monodisperse size distribution is a very strong assumption. Earlier LES modelling studies (Unterstrasser und Gierens 2010, Lewellen 2014, Lewellen, Meza und Huebsch 2014) have shown that only the largest ice crystals fall out from the contrail to form a fall streak, while the smaller ice crystals remain at the original altitude or fall only very slowly. I doubt that the claimed superiority of the numerical implementation matters as long as a monodisperse size distribution is assumed.

Allowing for a size distribution with pre-defined shapes would be one step forward. This can, however, lead to spurious sedimentation effects when ice mass and ice number vertical fluxes become decoupled. The bin approach (i.e. spectral microphysics) used by Lewellen (see citations above) or the Lagrangian approaches (Naiman, Lele und Jacobson 2011, Paoli, Hélie und Poinsot 2004, Unterstrasser, Gierens und Sölch, et al. 2017, Nybelen und Paoli 2009) are conceptually far more advanced than the present scheme. Back in 2006, (Huebsch und Lewellen 2006) demonstrated already that a monodisperse assumption is inadequate.

**3. Ice crystals habits**

Even after reading the text multiple times, I still cannot figure out how ice crystal habits are treated in your model. It remains fuzzy how you incorporate information about habits in your model.

- I do not understand what the habit percentages shown in Fig.6 are based on?
- Why should there be differences in young contrails based on background supersaturation?
- Why do the habit percentages change over time? Do you brute-force prescribe a habit depending on several atmospheric parameters? How is this backed up by observations?
- The observations may only provide an overall picture of habit classification (for example to include the bulk habit effect in radiative transfer applications as in Yang, 2010). However, I do not believe that those empirical relations should be used to predict the habit of individual contrail-cirrus lifecycles. Prescribing a habit transition for each contrail and for all ice crystals in a contrail is probably not meaningful.
- A change in habit percentage within a time period (e.g. during one time step) implies that a certain amount of ice crystals change from columnar to plate-like shape? Does this habit change override the non-isotropic growth of ice crystals described in Appendix E?

- Your habit formulations probably imply an increase in the aspect ratios of the ice crystals. Concerning, the difference between the two habit models, is it mostly an effect of more elongated ice crystal in your habit model? What other properties of your habit definitions (in addition to the aspect ratio) are relevant for your simulated contrail properties? How do your aspect ratios relate to other empirical studies defining habit types (Erfani und Mitchell 2016, Finlon, et al. 2019, Heymsfield, Lewis, et al. 2002)?
- Appendix E describes a non-isotropic growth of ice crystals. I cannot figure
  out how this relates to the habit definitions, which you seem to prescribe in a
  brute-force manner.

You apply the Chen & Lamb and Nelson & Baker parametrizations, which are poorly constrained in the temperature range, where most contrails form (around 220K). Extrapolating formulas that are valid only above 243 K to lower temperatures is, in my opinion, unreliable Temperatures of 243 K and higher belong to the mixed-phase cloud regime, whereas contrails generally form in the pure-ice cloud regime. Moreover, ice crystal habits can differ between natural cirrus and contrail cirrus.

Moreover, you state that discrepancies are largest below 233K. Contrails form typically only for temperatures below 225K.

I am not convinced that the papers cited in Appendix E are suited to describe the situation in contrails. In the end, you apply corrections that are not welljustified.

The statement that riming is omitted because of sparse data reveals a misinterpretation: riming requires supercooled droplets that splash onto ice crystals, but such droplets do not exist at contrail-forming temperatures.

In summary, given the many uncertainties in your approach, I am unsure whether including habits in the way you do makes the model better.

To be clear, I do not argue that treating ice crystals as spheres is reasonable. Several contrail LES models described in the literature incorporate ice crystal habits with increasing aspect ratios, but do not include any additional habit aspects that are extrapolated from a different temperature regime.

**4. Sedimentation**

How does your formulation of terminal velocity compare with other empirical fall-speed parameterisations (Mitchell 1996, Böhm 1989, McCorquodale und Westbrook 2021, Heymsfield und Westbrook 2010)?

I appreciate that you question the universal application of terminal velocity as the fall speed and that your model extends beyond this assumption.

However, I have strong doubts that sedimentation is adequately resolved in models that assume a monodisperse size distribution in each grid cell. Does contrail layering in your simulations resemble that of Fig. 4 in (Lewellen, Meza und Huebsch 2014)?

**5. Treatment of water vapour**

Relative humidity rises and water vapor becomes available when air masses ascend and cool adiabatically (or via other diabatic mechanisms). This is the typical driver of continued cloud growth. I am unaware of studies in which water vapor is replenished by vigorous vertical diffusion; in stratified flows vertical exchanges are usually weak. Could you provide evidence for the formula presented in line 458? Moreover, your formulation appears to suggest that the value of  $\alpha$  was selected to achieve a balanced state, yet the two processes need not be in equilibrium (and I doubt that vertical replenishment plays an important role).

**6. Odd 3D contrail initialization**

Initializing a contrail as a three-dimensional Gaussian plume is, in my view, inappropriate. For your type of application, a uniform contrail along flight direction would be more reasonable. In the cross-section perpendicular to the flight direction, a 2D Gaussian plume may be used, yet keeping in mind that this is already a strong simplification.

Currently, your Figs. 7 and 8 show a particular slice according to "We present the static contours at different times in the *x*–*z* plane along the track, i.e., along the center of the plume where maximum concentration is expected".

This sentence is ambiguous and raises more questions than it answers.

(unfortunately, such sentences occur throughout the manuscript and it is out of scope to mention all of them). What does the word "static" imply here? Why do the authors use the word "expect" instead of stating that the maximum concentration is determined? Do you pick the same slice for all times or does it jump from location to another? Is the displayed slice close to where the initial Gaussian plume has its maximum? It does not seem very robust to search for the maximum IWC as it can depend on unimportant turbulent fluctuations. It can be beneficial to pick the slice by evaluating the total ice mass over the x-z plane.

If the contrail is initialized uniformly along the flight direction, the results can be averaged over that direction, eliminating the need to choose a particular slice.

**7. Contrail upward movement**

(The following point could also be rated as a minor comment. But I classify it as major, since this manuscript aims at showcasing model capabilities and any possible model artifact should be clearly outlined).

Figures 12 and 13 show the contrail top rising by several hundred meters—a behavior that appears unphysical. While turbulence should not be so strong, sedimentation acts in opposite direction and radiative heating is absent in your simulations. What's the physical process behind this? Can you rule out numerical artifacts such as excessive numerical diffusion in your scheme?

**Minor comments**

- 1. Your separable ansatz seems to not work for scenarios with non-negligible vertical wind shear and slanted contrails. How severe is this restriction? Is it possible to simulate scenarios with vertical wind shear with your model?
- 2. Line 118: this sentence is not clear. What does vertical loss refer to, ice mass loss by sedimentation or a decrease in the contrail altitude? Small ambient updrafts are certainly not included in models to counteract sedimentation effects. They are the prerequisite of most cloud formation processes.
- 3. Line 170: number concentrations certainly drop due to dilution. But even the total ice crystal number is not conserved, as sedimentation and in-situ losses occur (Lewellen 2012, Lewellen 2014, Unterstrasser, Gierens und Sölch, et al. 2017).
- 4. I do not understand what is depicted in Fig. 4. Why does each panel contain three contrails? Why do your simulations start with contrails as broad as 30km? I cannot image which of your implemented physical processes lead to such a pronounced wavy pattern at later times?
- 5. I have never heard about diffusion blocking, loitering and preferential sweeping in the context of contrail evolution. These phenomena seem to be connected (at least partly) with the used numerical scheme. Unfortunately, I was not able to figure out whether these issues are also relevant to other contrail models.

**Technical comments**

- What's "necliated" at several occurrences?
- In Section 6.3, no information on the horizontal resolution is given, only vertical resolution is mentioned.
- Line 492: What's a "lack of water vapour budget"?
- Picking up my point, that information is not provided at ideal location: Appendix F
  presents the ice mass growth equation. Information about vertical profiles of
  thermodynamic conditions should be given elsewhere.

**Bibliography**

- Böhm, Hannes P. "A General Equation for the Terminal Fall Speed of Solid Hydrometeors." *J. Atmos. Sci.* 46 (1989): 2419–2427.
- Erfani, E., and D. L. Mitchell. "Developing and bounding ice particle mass- and area-dimension expressions for use in atmospheric models and remote sensing." *Atmos. Chem. Phys.* 16 (2016): 4379–4400.
- Finlon, J. A., et al. "A novel approach for characterizing the variability in mass–dimension relationships: results from MC3E." *Atmos. Chem. Phys.* 19 (2019): 3621–3643.
- Heymsfield, A. J., and C. D. Westbrook. "Advances in the estimation of ice particle fall speeds using laboratory and field measurements." *J. Atmos. Sci.* 67 (2010): 2469–2482.
- Heymsfield, A. J., et al. "A general approach for deriving the properties of cirrus and stratiform ice cloud particles." *J. Atmos. Sci.* 59 (2002): 3–29.

- Huebsch, W. W., and D. C. Lewellen. "Sensitivity Study on Contrail Evolution." *36 th AIAA Fluid Dynamics Conference and Exhibit*, AIAA 2006-3749 (2006): 1–14.
- Lewellen, D. C. "Analytic solutions for evolving size distributions of spherical crystals or droplets undergoing diffusional growth in different regimes." *J. Atmos. Sci.* 69 (2012): 417–434.
- Lewellen, D. C. "Persistent contrails and contrail cirrus. Part 2: Full Lifetime Behavior." *J. Atmos. Sci.*, 2014: 4420–4438.
- Lewellen, D. C., O. Meza, and W. W. Huebsch. "Persistent contrails and contrail cirrus. Part 1: Large-eddy simulations from inception to demise." *J. Atmos. Sci.* 71 (2014): 4399–4419.
- McCorquodale, M. W., and C. D. Westbrook. "TRAIL part 2: A comprehensive assessment of ice particle fall speed parametrisations." *Q. J. R. Meteorol. Soc.* 147 (2021): 605–626.
- Mitchell, D. L. "Use of mass- and area-dimensional power laws for determining precipitation particle terminal velocities." *J. Atmos. Sci.* 53, 12 (1996): 1710–1723.
- Naiman, A. D., S. K. Lele, and M. Z. Jacobson. "Large eddy simulations of contrail development: Sensitivity to initial and ambient conditions over first twenty minutes." *J. Geophys. Res.* 116 (2011): D21208.
- Nybelen, L., and R. Paoli. "Direct and large-eddy simulations of merging in co-rotating vortex system." *AIAA J.* 47 (2009): 157–167.
- Paoli, R., J. Hélie, and T. Poinsot. "Contrail formation in aircraft wakes." *J. Fluid Mech.* 502 (2004): 361–373.
- Unterstrasser, S., and K. Gierens. "Numerical simulations of contrail-to-cirrus transition Part 1: An extensive parametric study." *Atmos. Chem. Phys.* 10 (2010): 2017–2036.
- Unterstrasser, S., K. Gierens, I. Sölch, and M. Lainer. "Numerical simulations of homogeneously nucleated natural cirrus and contrail-cirrus. Part 1: How different are they?" *Meteorol. Z.* 26 (2017): 621–642.